# CoCoDiff: Correspondence-Consistent Diffusion Model for Fine-grained Style Transfer

**Wenbo Nie**[1,2*]   **Zixiang Li**[1,2*]   **Renshuai Tao**[1,2†]   **Bin Wu**[1,2]   **Yunchao Wei**[1,2]   **Yao Zhao**[1,2]

[1]Institute of Information Science, Beijing Jiaotong University
[2]Visual Intelligence + X International Joint Laboratory of the Ministry of Education

## Abstract

Transferring visual style between images while preserving semantic correspondence between similar objects remains a central challenge in computer vision. While existing methods have made great strides, most of them operate at global level but overlook region-wise and even pixel-wise semantic correspondence. To address this, we propose **CoCoDiff**, a novel *training-free* and *low-cost* style transfer framework that leverages pretrained latent diffusion models to achieve fine-grained, semantically consistent stylization. We identify that correspondence cues within generative diffusion models are under-explored and that content consistency across semantically matched regions is often neglected. CoCoDiff introduces a pixel-wise semantic correspondence module that mines intermediate diffusion features to construct a dense alignment map between content and style images. Furthermore, a cycle-consistency module then enforces structural and perceptual alignment across iterations, yielding object and region level stylization that preserves geometry and detail. Despite requiring no additional training or supervision, CoCoDiff delivers state-of-the-art visual quality and strong quantitative results, outperforming methods that rely on extra training or annotations. The source code is publicly available at: https://github.com/Wenbo-Nie/CoCoDiff.

## 1 Introduction

Diffusion models have achieved remarkable success in the field of generative artificial intelligence. By performing the diffusion process within a compressed latent space, Latent Diffusion Models (LDMs) Rombach et al. (2022) achieve substantial advantages in both generation efficiency and image quality. This efficient and flexible architecture has established them as a powerful backbone, including text-to-image generation Nichol et al. (2021); Ramesh et al. (2022); Saharia et al. (2022), image editing Meng et al. (2021); Tumanyan et al. (2022); Brooks et al. (2023), image restoration Lin et al. (2024); Wang et al. (2024b); Wu et al. (2024), and style transfer Wang et al. (2023; 2024a); Chung et al. (2024). Although diffusion models exhibit significant potential, style transfer remains challenging when semantic alignment and fine-grained correspondence are required. Beyond global appearance changes, high-fidelity stylization must respect region- and object-level structure. This calls for a principled way to mine the correspondence signals already encoded in pretrained diffusion models, enabling structure-aware, semantically consistent style transfer.

Neural style transfer Johnson et al. (2016); Zhang & Tang (2025); An et al. (2021) aims to render the content of one image in the visual appearance of another. A central challenge is to ensure that the stylistic features are applied in a semantically consistent manner, particularly across corresponding regions or objects Jiang & Chen (2025). With the rise of large-scale generative models, pre-trained latent diffusion models have become a powerful foundation for this task. However, as shown in Fig. 1(b), a representative method for training-free diffusion models suffers structural degradation and correspondence errors. In contrast, in Fig. 1(c), a representative of another neural-based method fails to capture the style information of the target image effectively. Similarly, Fig. 1(d) and (e),

---

*Equal contribution; †Corresponding author (email: *rstao@bjtu.edu.cn*).

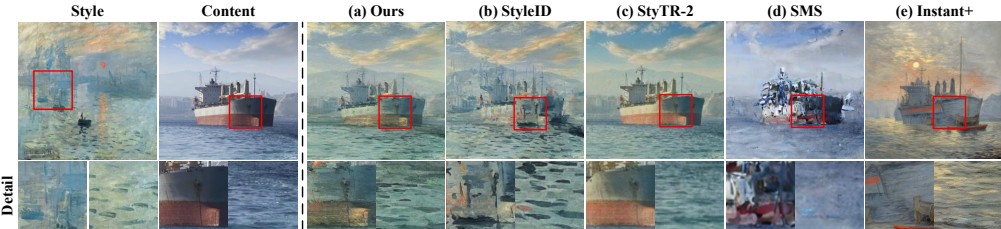

Figure 1: **Comparison of different style transfer methods.** We compare our proposed CoCoDiff with three other representative methods with zoomed-in details.

both prompt-guided methods, provided with a clear style prompt "oil painting, Claude Monet's Impression, Sunrise", also fail to capture the stylistic features. Almost all of these approaches treat the pre-trained model as a simple black-box generator. They either apply style globally(*e.g.*, StyleID) Chung et al. (2024); Deng et al. (2022), which leads to structural inconsistencies and content detail loss, or they build complex external modules that attempt to learn correspondences from scratch (*e.g.*, SMS Jiang et al. (2025)). Such methods are inefficient and overlook the powerful alignment information already present within the diffusion backbone itself. This results in a critical gap: a failure to directly mine and utilize the model's own semantic understanding for high-fidelity style transfer. Therefore, we argue that the full potential of these models for style transfer remains largely under-explored.

These issues point to a more fundamental, two-fold challenge at the core of style transfer. The first challenge is finding dense, semantically meaningful correspondences between content and style images. This is inherently difficult because content and style images often have vast differences in color, texture, and geometry. An effective method must identify features that are robust enough to match high-level concepts (*e.g.*, the structure of the ship, the texture of the waves in Fig. 1) across domains, yet precise enough to align fine-grained details (*e.g.*, the color of the sky, and the subtle, almost elusive representation of the ship in Fig. 1). The second, equally significant challenge is utilizing these correspondences to guide the stylization. Even with a perfect alignment map, how to perform feature injection is also a difficult problem. Therefore, a method is required to transfer local style characteristics onto the content's structure without creating disharmony in the overall global style. Solving this dual problem of correspondence-aware analysis and feature injection remains the primary barrier to achieving high-fidelity, structure-preserving style transfer.

To this end, we take a step forward by leveraging the implicit alignment capabilities of pre-trained diffusion models and introducing a cyclic optimization mechanism to enhance semantic consistency across domains. We propose Correspondence-Consistent Diffusion (**CoCoDiff**), a novel framework that achieves fine-grained feature correspondence and structure-preserving style transfer through adaptive, correspondence-aware guidance. Our approach is built upon the generative priors and semantic representation capabilities of pre-trained diffusion models, enabling structure-aware and semantically aligned stylization without the need for additional supervision or fine-tuning. Unlike existing methods that rely on coarse global matching or expensive supervised learning, CoCoDiff explicitly models pixel-level correspondences between content and style images by mining intermediate features at optimal denoising steps and network layers. Specifically, we perform a two-dimensional grid search over temporal and spatial resolutions of the diffusion backbone to identify the feature representations that best capture semantic structures. Using these features, we construct a dense correspondence map via cosine similarity, aligning each spatial location in the content image with its most semantically similar counterpart in the style image. Based on this alignment, we design a novel feature injection mechanism that selectively modulates the self-attention maps during the diffusion inversion process, transferring style information in a spatially aware manner. Furthermore, we introduce a cycle-consistency module that iteratively refines the stylized output by enforcing structural preservation and appearance fidelity across generations. The optimization process is guided by perceptual and style losses, as well as statistical alignment via Adaptive Instance Normalization (AdaIN), ensuring that the final stylized image remains faithful to the original content while accurately reflecting the reference style. Our contributions can be summarized as follows:

- We propose **CoCoDiff**, a training-free, diffusion-based framework for fine-grained, structure-preserving style transfer that operates directly on pretrained backbones without additional supervision or fine-tuning.

- We introduce a **pixel-wise semantic correspondence module** that mines intermediate features from pre-trained diffusion models to build dense alignment maps between content and style images, enabling region- and object-level stylization.

- We design a **cyclic optimization method** that integrates attention-guided feature injection with consistency constraints, enabling more stable and coherent stylization.

The remainder of the paper is organized as follows. Section 2 reviews prior work on diffusion models and style transfer. Section 3 presents the proposed CoCoDiff framework, including the fine-gained feature matching module and fitting cycle and iterative control strategy. In Section 4, we describe the experimental setup, datasets, evaluation metrics, and provide a comprehensive analysis of the results. Finally, Section 5 summarizes the contributions and discusses potential future directions.

## 2 RELATED WORK

### 2.1 DIFFUSION MODELS

Diffusion models have transformed generative modeling through iterative denoising, producing high-quality images, excelling in style transfer and similar applications. Starting with Denoising Diffusion Probabilistic Models (DDPM) Ho et al. (2020), achieving superior sample quality compared to GANs Goodfellow et al. (2014); Chen et al. (2016); Karras et al. (2019), with high computational cost. Denoising Diffusion Implicit Models (DDIM) Song et al. (2022) addressed this by using a non-Markovian sampling process, reducing inference steps. LDM Rombach et al. (2022), like Stable Diffusion, further improved efficiency by operating in a compressed latent space. The DALL·E2 Ramesh et al. (2022) and Imagen Saharia et al. (2022) integrate text-conditioned diffusion with CLIP Radford et al. (2021), enabling photorealistic text-to-image generation.

In downstream tasks, diffusion models have shown remarkable versatility. They are used for image editing through techniques like Null-Text Inversion Mokady et al. (2023) and ControlNet Zhang et al. (2023a), and DiffusionCLIP Kim et al. (2022). DIFT Tang et al. (2023) enables feature correspondence by aligning object-specific features across domains. For super-resolution, SRDiff Li et al. (2021) reconstructs high-quality images from low-resolution inputs, while RePaint Lugmayr et al. (2022) advances inpainting by seamlessly reconstructing missing regions. Additionally, style transfer is enhanced through approaches like DiffStyle Jiang & Chen (2025); Li (2024) which integrates text-guided diffusion with feature alignment for precise stylization. These advancements highlight diffusion models' fidelity and robustness across diverse generative tasks.

### 2.2 STYLE TRANSFER

The field of neural style transfer has evolved significantly since the seminal work of Gatys et al. Gatys et al. (2016), who showed that hierarchical layers in CNNs can separate content structures from style textures, more efficient approaches like including AdaIN Huang & Belongie (2017) and WCT Li et al. (2017) soon followed. To better preserve structural details, researchers introduced transformer architectures, StyTR$^2$ Deng et al. (2022) pioneered the first step and StyleFormer Wu et al. (2021) incorporated transformer modules into CNN pipelines. Moreover, patch-based methods Liao et al. (2017); Shang et al. (2025); Wang et al. (2022); Chen & Schmidt (2016); Li & Wand (2016); Sheng et al. (2018) provide fine-grained local constraints for style transfer and offer important insights into maintaining style stability for objects appearing at different positions or scales. Diffusion models have revolutionized this field through two main approaches: training-based methods like StyleDiffusion Wang et al. (2023); Li (2024), use neural flows, while training-free approaches like DiffArtist Jiang & Chen (2025); Chung et al. (2024); Xu et al. (2024), manipulate attention in pre-trained models. Recent innovations include InstantStyle-Plus Wang et al. (2024a), which balances content and style using inverted noise, and FreeStyle He et al. (2024) uses a dual-stream encoder for text-guided transfer. However, these methods have largely overlooked the challenge of maintaining style consistency for identical objects in style transfer.

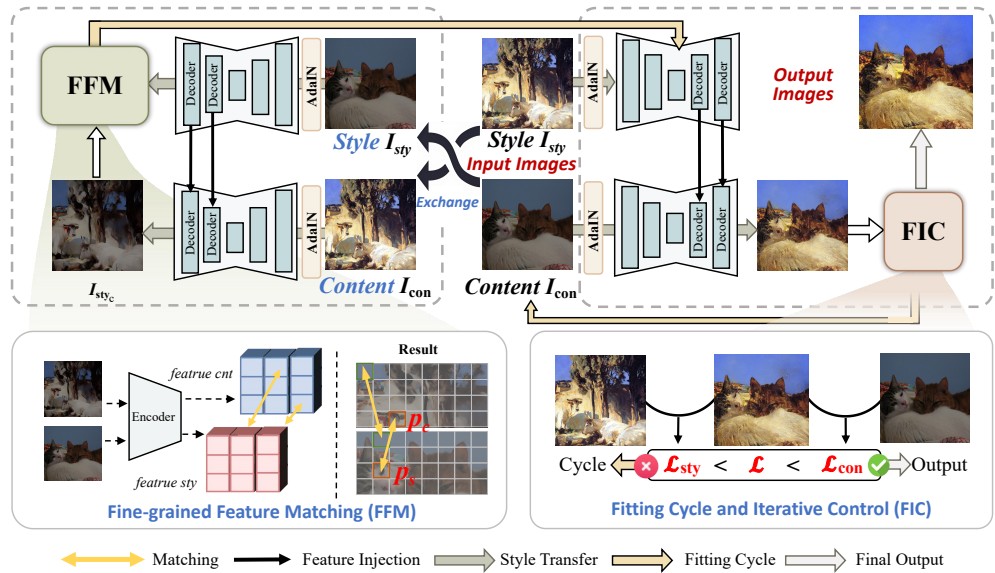

Figure 2: Framework Overview of our proposed Correspondence-Consistent Diffusion (CoCoDiff).

## 3 METHODOLOGY

### 3.1 PRELIMINARY

Starting from pure noise $x_T \sim \mathcal{N}(\mathbf{0}, \mathbf{I})$, the diffusion model predicts noise $\epsilon_\theta(x_t, t)$ at each timestep $t$ and updates $x_t$ to $x_{t-1}$ via:

$$q(\mathbf{x}_t \mid \mathbf{x}_{t-1}) = \mathcal{N}\left(\mathbf{x}_t; \sqrt{1 - \beta_t}\, \mathbf{x}_{t-1}, \beta_t \mathbf{I}\right) \tag{1}$$

where $\mathbf{x}_t$ and $\mathbf{x}_{t-1}$ are the latent variables at timestep $t$ and $t-1$ with $t \in \{1, \dots, T\}$. The parameter $\beta_t \in (0, 1)$ defines the noise variance at each timestep, as determined by a predefined schedule $\beta_t$.

Stable Diffusion (SD) has a compressed latent space. It uses an encoder to transform input images into a compact latent representation, and a decoder to reconstruct them. The U-Net architecture of Stable Diffusion generates images from latent representations through iterative denoising. A key point of this process is self-attention (SA) blocks. For a feature $\phi$, the SA block computes as:

$$\phi^{\text{out}} = \text{Attn}(Q, K, V) = \text{softmax}\left(\frac{QK^T}{\sqrt{d}}\right) \cdot V. \tag{2}$$

In our work, we perform style transfer by leveraging the self-attention (SA) mechanism of Stable Diffusion (SD) to inject features from a style image $I_s$ into a content image $I_c$. We adopt the DDIM inversion procedure Rombach et al. (2022) to traverse the denoising trajectory from $t=0$ (clean image) to $t=T$ (Gaussian noise). At each timestep $t$, we extract queries $Q_c^t$ from $I_c$ and keys and values $K_s^t, V_s^t$ from $I_s$. Stylization is effected by replacing the content keys/values $K_c^t, V_c^t$ with $K_s^t, V_s^t$ inside the SA blocks, and computing:

$$\phi_c^{\text{out}} = \text{Attn}(Q_c^t, K_s^t, V_s^t). \tag{3}$$

This process forms the foundation for injecting style features.

### 3.2 FRAMEWORK OVERVIEW

As illustrated in Fig. 2 and Algorithm 1, our method consists of two key stages: *Fine-grained Feature Matching* that mines correspondences of content and style; *Fitting Cycle and Iteration Control* that ensures consistency in the generative process. We will detail them at the following part.

### 3.3 FINE-GRAINED FEATURE MATCHING

**Diffusion features for correspondence.** Pretrained diffusion U-Nets have the ability to encode semantic cues across timesteps $t$ and network layers $l$ Tang et al. (2023). We treat intermediate activations as per-pixel descriptors for dense correspondence. For notational simplicity, throughout this subsection we use $I_c(p_c)$ and $I_s(p_s)$ to denote the pixels of content and style images, respectively.

**Dense correspondence by diffusion features.** Just as a painter maintains stylistic consistency when depicting the same object, our method prioritizes stylistic coherence across semantically corresponding regions to ensure entire visual consistency. Exploiting the semantic encoding of diffusion features, we extract feature maps and, for each spatial location $p_c$ in the content image, identify the most semantically similar location $p_s$ in the style image through cosine similarity measurements of normalized diffusion features:

$$\cos(p_c, p_s) \;=\; \frac{I_c(p_c) \cdot I_s(p_s)}{\|I_c(p_c)\| \, \|I_s(p_s)\|}. \tag{4}$$

This process yields a dense semantic correspondence map that guides style transfer with improved accuracy and structure preservation.

**Selecting optimal combination.** We employ a two-dimensional grid search to determine the optimal combination of timestep $t$ and network layer $l$, aiming to balance semantic representation and low-level detail. The optimal pair $(t^*, l^*)$ is selected by maximizing a correspondence quality metric $\mathcal{M}(t, l)$ over predefined candidate sets $\mathcal{T}$ and $\mathcal{L}$:

$$(t^*, l^*) = \arg_{t,l} \max_{t \in \mathcal{T}, \, l \in \mathcal{L}} \mathcal{M}(t, l). \tag{5}$$

Here, $\mathcal{M}(t, l)$ evaluates the alignment quality based on the extracted feature maps at timestep $t$ and layer $l$. To clarify how $\mathcal{M}(t, l)$ is defined and quantified, we follow the standard semantic correspondence protocol and extract diffusion features from image pairs on benchmark datasets such as SPair-71k Min et al. (2019). For each content keypoint, we identify the location in the style image that yields the highest cosine similarity, and adopt the Percentage of Correct Keypoints (PCK) as the evaluation criterion to determine whether the predicted correspondence falls within an acceptable distance of the ground-truth keypoint. The metric $\mathcal{M}(t, l)$ is thus defined as the average PCK score over all evaluated samples, reflecting the reliability of semantic correspondence within the feature space $(t, l)$. Higher values indicate that the feature configuration provides a more reliable semantic alignment foundation for subsequent style transfer.

We fix $(t^*, l^*)$ and use the resulting correspondence map to steer correspondence-aware feature injection in the subsequent feature injection stage. The final spatial location $p_s^*$ is:

$$p_s^* = \arg_{p_s} \max_{p_s \in I_s} \cos(p_c, p_s). \tag{6}$$

### 3.4 FITTING CYCLE AND ITERATIVE CONTROL

We then use the semantic correspondence map mentioned before to guide style injection. To further enhance the semantic consistency of the feature maps, we adjust attention weights based on the cosine similarity at corresponding positions $p_s^*$ in $I_s$:

$$feat[k] = w \cdot attn[k][p_s^*] + feat[k][p_c], \tag{7}$$

where $k$ denotes the feature channel, $p_c$ is the spatial location in content image, $p_s^*$ is the corresponding location in stylized content image and $w$ is a weighting factor controlling the contribution of the attention features. This update mechanism ensures precise injection of style features into semantically aligned regions while maximally preserving the structural information of the content image, resulting in high-quality style transfer. However, feature-based stylization is fragile when the statistics of the style and content images vary widely. Directly matching features across these disparate distributions can lead to imprecise or semantically misaligned results.

**Closed-loop refinement.** To progressively enhance correspondence while preserving content structure and improving visual fidelity, we introduce a cycle-consistent refinement with closed-loop feature fusion. In each fitting cycle, we first identify a set of images $I_c$ and $I_s$. We use the previously

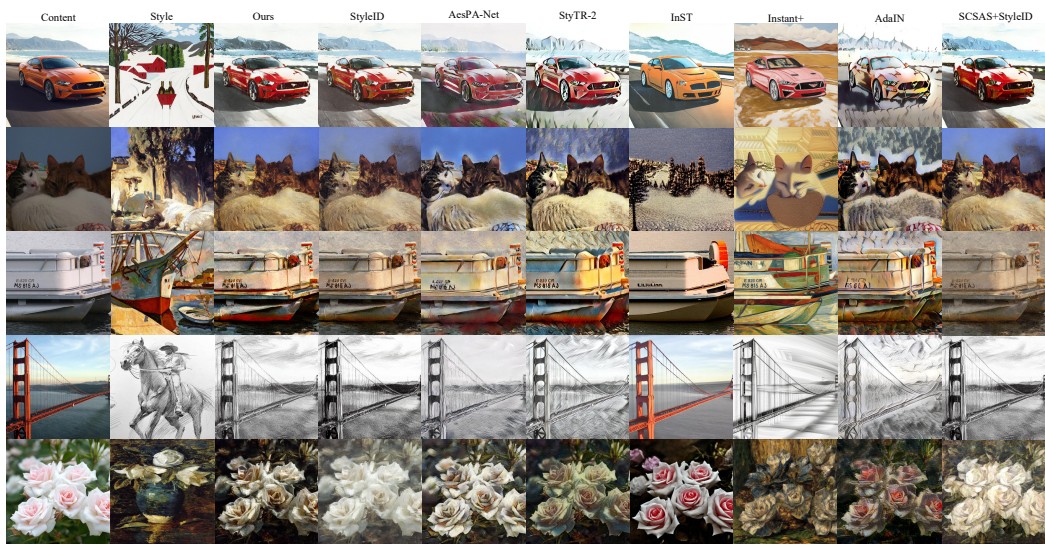

Figure 3: Qualitative comparison. We compare **CoCoDiff** (Ours) with seven representative methods, selected from diffusion-based, patch-based, CNN-based, transformer-based, and other approaches, to provide a comprehensive evaluation.

---

**Algorithm 1:** Correspondence-Guided Style Transfer

---

**Input:** $I_c$: content image; $I_s$: style image; $I_{sty_c}$: style image with content features; $f_\theta$: pretrained diffusion U-Net; $\mathcal{T}$: candidate timesteps; $\mathcal{L}$: candidate layers; $w$: injection weight; $\tau_c, \tau_s$: stopping thresholds; $z$: max iterations

**Output:** $I_{\text{gen}}^{(z^*)}$

```
// Stage A: correspondence
```
$I_{sty_c} \leftarrow \text{Attn}(Q_s, K_c, V_c)$          `// equation 3`

Extract $\{F_c^{t,l}\}, \{F_{sty_c}^{t,l}\}$ for $t \in \mathcal{T}$, $l \in \mathcal{L}$; $(t^*, l^*) = \arg\max \mathcal{M}(t,l)$    `// equation 5`

for each $p_c$: $p_{sty_c}^* = \arg\max_{p_{sty_c}} \cos\left(F_c^{t^*,l^*}(p_c), F_{sty_c}^{t^*,l^*}(p_{sty_c})\right)$   `// equation 4, equation 6`

```
// Stage B: fitting & control
```
**for** $z = 1$ **to** $Z$ **do**

    $y_{cs} = \sigma(y_s)\frac{y_c - \mu(y_c)}{\sigma(y_c)} + \mu(y_s)$          `// equation 11`

    $feat[k] \leftarrow w \cdot attn[k][p_{\text{sty}_c}^*] + feat[k][p_c]$          `// equation 8`

    $\mathcal{L}_{content} = \|\text{Sobel}(I_{\text{gen}}^{(z)}) - \text{Sobel}(I_c)\|$, $\mathcal{L}_{style} = \sum_l \|G(I_{\text{gen}}^{(z),l}) - G(I_s)\|_F^2$   `// equation 10`

    **if** $\mathcal{L}_{content} > \tau_c$ **and** $\mathcal{L}_{style} < \tau_s$ **then**

        $\llcorner$ **break**

**return** $I_{\text{gen}}^{(z)}$

---

mentioned Feature Injection method to enable the style image to learn the style features of the content image, updating its attention map to obtain the reverse style transfer image $I_{sty_c}$. Then, using the Feature Matching method, we build correspondences between $I_c$ and $I_{sty_c}$ to obtain coordinate pairs $(p_c, p_{sty_c}^*)$. For $I_c$ and $I_s$, we perform feature injection again, leveraging the coordinate pairs to inject attention values of corresponding features, achieving a correspondence-consistent cycle. The updated formulation is as follows:

$$feat[k] = w \cdot attn[k][p_{sty_c}^*] + feat[k][p_c]. \tag{8}$$

**Optimal objectives.** In each iteration, we execute one fitting cycle to progressively optimize the generated image. Specifically, after each iterative generation, we update the feature map via Eq. 8, achieving semantically consistent fusion of style and content features. To evaluate the quality of the feature map generated in each cycle and enable adaptive iterative optimization, the following objective functions are to measure content and style fidelity concerning two aspects. Here, the image generated at iteration $z$ is denoted as $I_{gen}^{(z)}$. The content perceptual loss $\mathcal{L}_{content}$ is defined as:

$$\mathcal{L}_{content} = \|\text{Sobel}(I_{gen}^{(z)}) - \text{Sobel}(I_c)\|, \tag{9}$$

where Sobel$(\cdot)$ computes edge maps to capture structural information. Meanwhile, the style perceptual loss $\mathcal{L}_{style}$ can be defined as:

$$\mathcal{L}_{style} = \sum_{l \in \text{layers}} \|G(I_{gen}^{(z),l}) - G(I_s)\|_F^2, \tag{10}$$

where $G$ denotes the Gram Gatys et al. (2015; 2016) matrix capturing textural style properties across selected feature layers.

The iterative process terminates when both $\mathcal{L}_{content} > \tau_c$ and $\mathcal{L}_{style} < \tau_s$, where $\tau_c$ and $\tau_s$ are predefined thresholds. This ensures the generated image prevents over-stylization or structural distortion. We describe in detail how we choose these hyperparameters in the next section.

**Tone harmonization.** In the process of artistic style transfer, the harmonization of color and tone plays a pivotal role in achieving effective stylization. To this end, we employ AdaIN Huang & Belongie (2017) to modulate statistical information during the style transfer process. Specifically, we leverage the latent variables of the content image $y_c$ and style image $y_s$, to facilitate tone transfer through statistical alignment, as expressed in the following formula:

$$y_{cs} = \sigma(y_s) \frac{y_c - \mu(y_c)}{\sigma(y_c)} + \mu(y_s), \tag{11}$$

where $\mu(\cdot)$ and $\sigma(\cdot)$ denote the channel-wise mean and standard deviation, respectively. This approach ensures that the tone information of the style image is effectively integrated while preserving the structural integrity of the content image, thereby enhancing the quality of the stylized output.

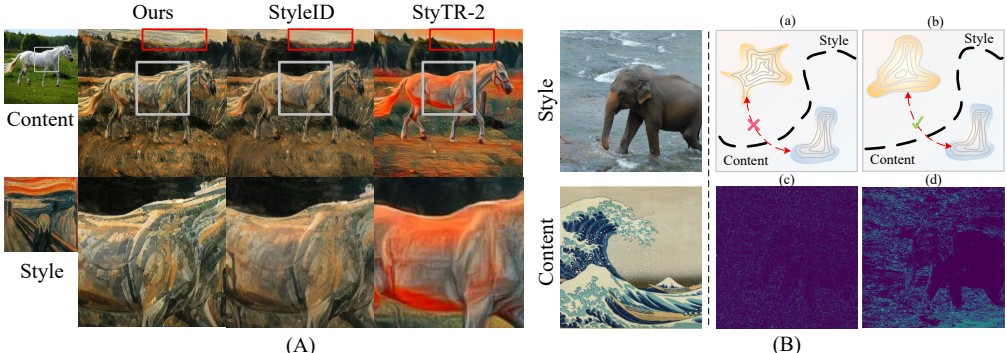

Figure 4: **(A) Qualitative comparison with additional zoomed-in details**. We compare our method with StyleID and StyTR$^2$ as baseline approaches, highlighting the differences through zoomed-in details. **(B) The illustration of cycle-based image style transfer.** (a) Direct feature matching between the style image and the content image often results in low matching accuracy and feature correspondence failure. (b) By first transforming the style image to adopt the content image's style before performing feature matching, the matching accuracy is significantly improved. (c) Direct correspondence result. (d) Indirect correspondence result.

## 4 EXPERIMENTS

**Datasets and Settings** We use MS-COCO Lin et al. (2015) as the content dataset and WikiArt Phillips & Mackintosh (2011) as the style dataset. We conduct all experiments on a single NVIDIA RTX 4090 GPU using the pre-trained Stable Diffusion V1.4 model as it allows for fair comparison, with DDIM Rombach et al. (2022) sampling over 50 timesteps ($t = \{1, \ldots, 50\}$) and the attention temperature scaling parameter $\gamma = 0.7$.

**Evaluation Protocol** We compare our method against nine style transfer methods: StyleID Chung et al. (2024), AesPA-Net Hong et al. (2023), StyTR$^2$ Deng et al. (2022), InST Zhang et al. (2023b), InstantStyle-Plus Wang et al. (2024a), AdaIN Huang & Belongie (2017), SCSA Shang et al. (2025),FreeStyle He et al. (2024), SMS Jiang et al. (2025). Experiments are conducted by configuring these methods according to their publicly available codes and settings. To quantitatively evaluate the style transfer model, we conduct experiments across 13 distinct artistic styles, including oil painting, kids' illustration, watercolor, Ghibli, landscape woodblock printing, etc. For quantitative comparison purposes, we adopt a methodology similar to StyleID Chung et al. (2024) and StyTR$^2$ Deng

| Metric | Ours | StyleID | AesPA | StyTR$^2$ | InST | Instant+ | AdaIN | SCSA | FreeStyle | SMS |
|---|---|---|---|---|---|---|---|---|---|---|
| FID ↓ | **18.432** | 21.010 | 19.645 | 18.886 | 21.541 | 20.982 | 18.672 | 20.835 | 23.654 | 31.266 |
| LPIPS ↓ | **0.549** | 0.565 | 0.556 | 0.587 | 0.785 | 0.584 | 0.612 | 0.562 | 0.689 | 0.821 |
| ArtFID ↓ | **30.100** | 34.446 | 32.124 | 31.559 | 40.235 | 34.820 | 31.711 | 34.106 | 41.640 | 58.756 |
| CFSD ↓ | **0.609** | 0.619 | 0.632 | 0.687 | 0.881 | 0.710 | 0.642 | 0.612 | 0.660 | 0.704 |

Table 1: **Quantitative evaluation results.** We compare every methods across multiple metrics. Columns $2^{nd}$-$9^{th}$ are reference-guided methods while columns $10^{th}$-$11^{th}$ are prompt-based methods.

et al. (2022), randomly selecting 30 content images and 30 style images from each dataset to create a comprehensive evaluation suite. We employ four established metrics: FID Heusel et al. (2018), LPIPS Zhang et al. (2018), ArtFID Wright & Ommer (2022), and CFSD Chung et al. (2024). Lower FID scores indicate greater similarity between stylized images and the target style domain. LPIPS quantifies perceptual similarity between stylized and original images. ArtFID measures alignment with human aesthetic preferences, with lower values reflecting superior performance. CFSD is a content-focused metric that evaluates the preservation of the original image's structural integrity. Together, these metrics provide a comprehensive evaluation of style transfer quality and content preservation.

## 4.1 QUANTITATIVE COMPARISON

Table 1 reports the quantitative comparison on four metrics: FID, LPIPS, ArtFID, and CFSD (↓ represents lower is better). Our method showcases a comprehensive quantitative comparison with baseline approaches, highlighting its superior efficacy in striking a robust balance between vivid style transfer quality and content preservation.

## 4.2 QUALITATIVE COMPARISON

As shown in Fig. 3, CoCoDiff achieves superior performance in style transfer, producing visually compelling results with precise feature alignment. For example, in the first row, the car exhibits consistent and vibrant coloring, while the background mountains acquire deeper hues. Similarly, in the third row, the boat's hull seamlessly shifts to the target red color. These results highlight our approach's ability to achieve fine-grained stylization while preserving content integrity, significantly outperforming baseline methods in both style expressiveness and visual coherence. Additionally, we provide a qualitative comparison with zoomed-in details in Fig. 4(A). For the painting *The Scream*, characterized by its distorted forms, vibrant colors, and dynamic lines, CoCoDiff more effectively preserves stylistic fluidity and intricate details compared to other approaches. This reinforces the precision of our method in achieving advanced feature alignment and stylistic fidelity.

## 4.3 DIFFUSION MODEL SCALING AND CORRESPONDENCE PERFORMANCE

In order to demonstrate the model-agnostic nature of CoCoDiff, we conduct experiments using SD v1.4, SD v1.5, SD v2.1, and SDXL with identical configurations, as shown in the Tab. 2. The results indicate that CoCoDiff adapts well to different scales of diffusion models, delivering stable performance. However, significant differences arise in terms of semantic consistency and fine-grained style transfer across these models. While SDXL shows strong performance in style transfer quality, its stability in fine-grained semantic alignment decreases. This is due to the need for fine-tuning hyperparameters such as $\gamma$ and temperature for different models to optimize visual performance.

| Metric | SD v1.4 | SD v1.5 | SD v2.1 | SD XL |
|---|---|---|---|---|
| FID ↓ | 18.432 | 17.995 | 18.252 | 21.412 |
| LPIPS ↓ | 0.549 | 0.525 | 0.621 | 0.635 |
| ArtFID↓ | 30.1 | 28.94 | 31.18 | 36.69 |
| CFSD ↓ | 0.609 | 0.589 | 0.766 | 0.714 |

Table 2: Comparison across different diffusion models.

| Sobel | Gram | FID | LPIPS | CFSD |
|---|---|---|---|---|
| - | - | 26.513 | 0.697 | 0.804 |
| ✓ | - | 29.845 | 0.506 | 0.631 |
| - | ✓ | 23.471 | 0.753 | 0.761 |
| ✓ | ✓ | 18.432 | 0.549 | 0.609 |

Table 3: Results for Sobel and Gram combinations.

### 4.4 EFFECTIVENESS OF FEATURE EXTRACTION

To demonstrate the critical limitations, we present our comparative results in Fig. 4(B). The left side of the figure displays the original content image and the target style image, which share similar wave patterns. We perform style transfer using three methods: a baseline direct style transfer method, a direct correspondence method, and our indirect correspondence method, which relies on $I_{sty_c}$.

Difference heatmaps are obtained by subtracting the results of the two correspondence methods from the baseline output; brighter colors indicate more pronounced differences. As shown in Fig. 4(B.c), the direct correspondence method produces global differences, revealing its failure to maintain a consistent stylized output compared to the baseline. In contrast, the heatmap (B.d) for our indirect method shows strong differences localized to the wave patterns, indicating that it successfully preserves the elephant's contour while applying a high-fidelity style transfer. These observations underscore the inadequacy of a direct approach and validates our indirect feature-based strategy.

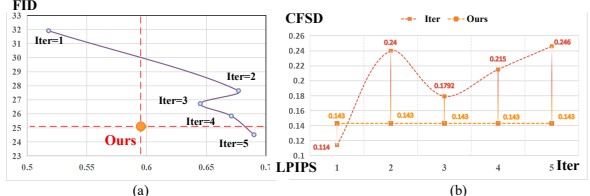

Figure 5: **Quantitative comparison of the cycle module.** (a) Balance between LPIPS and FID metrics across iterations. (b) CFSD variations across iterations.

### 4.5 EFFECTIVENESS OF CYCLE MODULE

Recognizing the limitations of fixed iteration counts in style transfer, we investigate the crucial role of adaptive iteration within our CoCoDiff method. We conduct experiments comparing style transfer across 10 content and 10 style images, evaluating fixed iterations (ranging from 1 to 5) against an adaptive gating strategy with early stopping. Performance evaluation employs FID, CFSD, and LPIPS. As shown in Fig. 5(a), our adaptive gating approach achieves an optimal tradeoff between style transfer and content

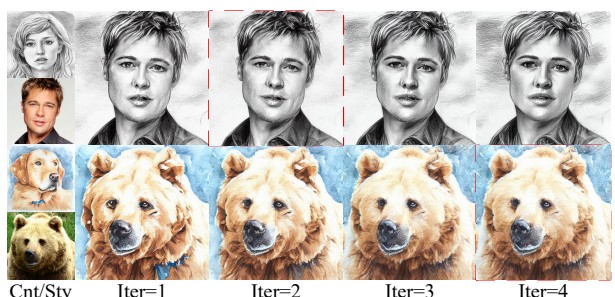

Figure 6: **Visual results of the iteration process.** The group with the best quality is highlighted by a red line.

preservation, significantly surpassing fixed iteration counts in both LPIPS and FID metrics. Fig. 5(b) further confirms this advantage through CFSD scores, demonstrating content preservation outcomes. Additionally, Fig. 6 presents two sets of visualized iteration results, showcasing optimal visual quality with a red line for adaptive iteration selection.

### 4.6 ABLATION STUDY

To optimize the attention injection weight $w$, we introduce a scaling factor to modulate the intensity of style feature injection based on cosine similarity. We conduct an ablation study, evaluating the visual quality of the generated images with $w$ set to 0.3, 0.6, 1.8, 2.4 and 3.0. The results are presented in the Tab 4a. The results indicate that $w = 0.6$ strikes an optimal balance between content preservation and style fidelity, maintaining semantic consistency and structural integrity. Lower values (*e.g.*, $w = 0.3$) may lead to insufficient stylization, while higher values (*e.g.*, $w = 3.0$) risk introducing structural distortions. These results underscore the critical role of carefully tuning $w$ to achieve high-quality, visually coherent style transfer outcomes. Tab. 4b validates AdaIN Huang & Belongie (2017)'s effectiveness in harmonizing color and tone during the style transfer process.

Table 3 reports an ablation on the Sobel and Gram. Without either component, the model produces the weakest results across all metrics. Adding Sobel alone significantly improves LPIPS and CFSD by enforcing clearer structural alignment, while adding Gram alone yields better FID by enhancing global style coherence but tends to distort local perceptual details. When combined, Sobel and Gram provide complementary benefits, achieving the best overall performance with the lowest FID

| $w$ | 0.3 | 0.6 | 1.8 | 2.4 | 3 |
|---|---|---|---|---|---|
| FID | 23.717 | **18.432** | 21.911 | 33.980 | 35.937 |
| LPIPS | 0.681 | **0.549** | 0.588 | 0.656 | 0.667 |
| CFSD | 0.642 | **0.609** | 0.746 | 0.735 | 0.791 |

|  | w/ AdaIN | w/o AdaIN |
|---|---|---|
| FID | **18.432** | 20.515 |
| LPIPS | **0.549** | 0.762 |
| CFSD | **0.609** | 0.646 |

(a) Effect of feature injection weight ($w$).     (b) Effect of AdaIN on model performance metrics.

Table 4: Ablation study analyzing the impact of various design choices on model performance.

and strongest content–style balance. This demonstrates that integrating structural cues with style correlation statistics leads to more stable optimization and higher-quality stylization.

## 4.7 USER STUDY

We conduct a user study to evaluate the performance of our proposed style transfer method in terms of semantic consistency and style fidelity. Participants view content and style images alongside randomized style transfer results from various methods to ensure unbiased comparisons.

We collected comparisons from 25 participants (aged 19–45) across 10 distinct content images and 8 style images. Table 5 summarizes the preference rates, with CoCoDiff achieving the highest scores across all criteria, thereby substantiating its effectiveness. These findings not only further substantiate the efficacy of our approach in producing high-quality stylizations but also provide compelling evidence of a consistent user preference for results that maintain strong feature alignment while exhibiting expressive and visually appealing stylistic characteristics.

| Method | Style | Content | Avg |
|---|---|---|---|
| StyleID  Chung et al. (2024) | 0.212 | 0.135 | 0.174 |
| AesPA  Hong et al. (2023) | 0.033 | 0.134 | 0.084 |
| StyTR$^2$  Deng et al. (2022) | 0.132 | 0.036 | 0.084 |
| InST  Zhang et al. (2023b) | 0.004 | 0.025 | 0.015 |
| Instant+  Wang et al. (2024a) | 0.002 | 0.174 | 0.088 |
| AdaIN  Huang & Belongie (2017) | 0.038 | 0.013 | 0.026 |
| FreeStyle  He et al. (2024) | 0.021 | 0.013 | 0.017 |
| SMS  Jiang et al. (2025) | 0.012 | 0.032 | 0.022 |
| CoCoDiff(Ours) | **0.546** | **0.438** | **0.492** |

Table 5: User study results in consistency and quality.

## 5 CONCLUSIONS

In this paper, we introduce Correspondence-Consistent Diffusion (CoCoDiff), a novel training-free framework that utilizes pre-trained latent diffusion models to achieve high-fidelity style transfer. By extracting intermediate features to establish pixel-wise correspondences and applying cyclic optimization techniques, CoCoDiff ensures robust semantic consistency and preserves structural integrity between the content and the transferred style. Through extensive experiments across various benchmarks, we demonstrate that CoCoDiff significantly outperforms current state-of-the-art models, not only in terms of style fidelity but also in content alignment. These results reveal the potential of CoCoDiff to unlock new possibilities for diffusion-based generative tasks, paving the way for more effective and flexible style transfer solutions in generative modeling.

## 6 ACKNOWLEDGE

This work was supported by National Natural Science Foundation of China (No. 62506030, U24B20179, 92470203), Beijing Natural Science Foundation (No. L242021, L242022) and the Fundamental Research Funds for the Central Universities (2024XKRC082).

## ETHICS STATEMENT

This work focuses on developing a training-free style transfer framework, CoCoDiff, built upon pre-trained diffusion models. Our research does not involve the collection of personal data, human subjects, or sensitive information, and all datasets used are publicly available under appropriate licenses. We encourage responsible use of CoCoDiff within creative, educational, and research contexts, and emphasize that any deployment of this method should adhere to ethical guidelines

and legal standards. We hope our work can inspire further research and contribute to advancing the positive impact of generative modeling in both academic and real-world contexts.

## REPRODUCIBILITY STATEMENT

The code is available in the supplementary materials. For full reproducibility, we have detailed datasets used for testing are also provided as described in Sec. 4 and Appendix D.1. The experimental hyperparameters and model selections in the Sec. 4 and the Appendix D.2.

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

## A  OVERVIEW

This supplementary material supports the main paper with:

- The use of Large Language Models (Section  B)

- Evaluation metrics (Section  C).

- Experiment details (Section  D).

- Visual comparisons results (Section  F).

- Impact (Section  G).

## B  USE OF LLMs

In accordance with ICLR 2026 policy on AI assistance disclosure, we acknowledge the use of large language models in our paper preparation process. Our usage limits to language polishing and grammatical improvements of the final manuscript. The language models do not involve experimental design, data analysis, result interpretation, or the generation of substantive content. They serve solely as writing assistance tools to improve clarity and readability of text already authored by the human authors listed on this paper. The scientific contributions, methodology, experiments, results, and conclusions belong entirely to the work of the human authors.

## C  EVALUATION METRICS

### C.1  LPIPS

LPIPS Zhang et al. (2018) is a perceptual metric designed to mimic how humans perceive image differences. Instead of comparing pixels directly, it measures the distance between images in the feature space of a deep network that has been trained on a perceptual similarity task. To calculate LPIPS, a reference image and a generated image are fed into a pre-trained network, and feature maps are extracted from several of its layers. These feature maps are L2-normalized, and a weighted L2 distance is computed between the corresponding features of the two images at each layer. The final LPIPS score is the sum of these weighted distances. The formula is given by:

$$d(x, x_0) = \sum_l \frac{1}{H_l W_l} \sum_{h,w} w_l \cdot \|\phi_l(x)_{h,w} - \phi_l(x_0)_{h,w}\|_2^2. \tag{12}$$

A lower LPIPS score indicates higher perceptual similarity between the generated and real images, making it a valuable metric for tasks like image reconstruction and super-resolution.

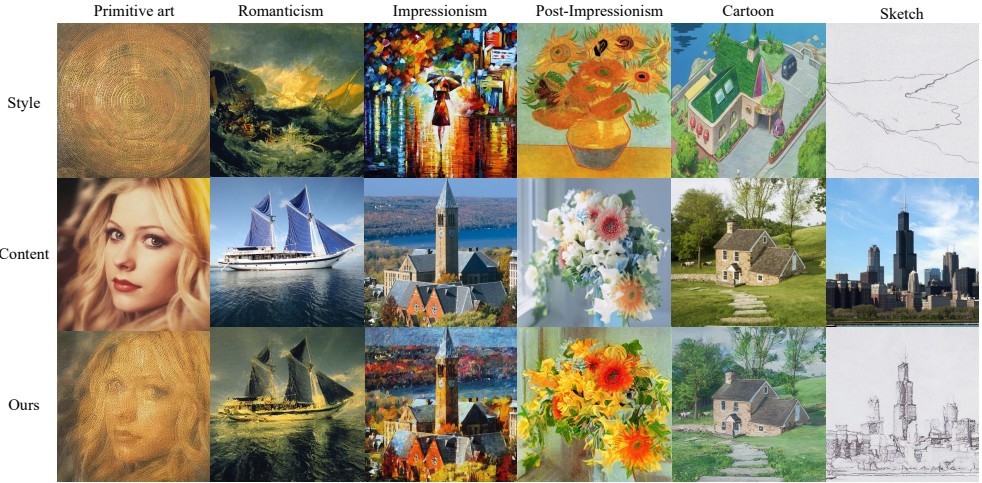

Figure 7: Additional classic visual outcomes in diverse artistic styles.

## C.2 FID

FID Heusel et al. (2018) assesses the quality and realism of an entire set of generated images from a statistical perspective. It doesn't compare individual images but rather measures the distance between the feature distributions of a generated image set and a real image set. The calculation relies on a pre-trained Inception-v3 network. Feature vectors are extracted from the last pooling layer for both the real and generated image sets. The mean vectors ($\mu$) and covariance matrices ($\Sigma$) are then computed for both sets. FID is the Fréchet distance between these two multivariate Gaussian distributions, calculated using the formula:

$$FID = \|\mu_r - \mu_g\|_2^2 + \mathrm{Tr}(\Sigma_r + \Sigma_g - 2(\Sigma_r\Sigma_g)^{1/2}). \tag{13}$$

A lower FID score indicates that the generated image distribution is closer to the real image distribution, which implies better diversity and realism. FID is widely considered the standard metric for evaluating Generative Adversarial Networks (GANs).

## C.3 ARTFID

ArtFID Wright & Ommer (2022) is a composite metric specifically designed for evaluating neural style transfer models, aiming to balance content fidelity and style fidelity in a way that aligns better with human subjective judgments. It combines LPIPS, which measures perceptual content similarity, with FID, which assesses style distribution realism, through a multiplicative formula to penalize deviations in either aspect. To calculate ArtFID, first compute the average LPIPS score between the stylized images and the original content images, and the FID score between the stylized images and the reference style images, using pre-trained networks like VGG for LPIPS and Inception-v3 for FID. The final ArtFID score is then obtained by the formula:

$$\mathrm{ArtFID} = (1 + \mathrm{LPIPS}) \times (1 + \mathrm{FID}). \tag{14}$$

A lower ArtFID score indicates superior style transfer performance, where both content preservation and style matching are optimized, making it particularly useful for comparing methods in artistic image generation tasks.

## C.4 CFSD

CFSD Chung et al. (2024) is a content-focused metric that evaluates the structural similarity in neural style transfer by emphasizing spatial relationships between image patches, addressing limitations in metrics like LPIPS that may be influenced by style elements. It operates on feature maps

extracted from a pre-trained VGG19 network's conv3 layer to capture mid-level structural details. To calculate CFSD, extract feature maps $F$ from both the content image $I_c$ and stylized image $I_{cs}$, compute self-correlation matrices $M = F \times F^T$, and normalize each row via softmax to form probability distributions $S$. The CFSD score is the average Kullback-Leibler divergence across these distributions:

$$\text{CFSD} = \frac{1}{hw} \sum_{i=1}^{hw} D_{\text{KL}}(S_{c_i} \| S_{cs_i}). \tag{15}$$

A lower CFSD score signifies better preservation of the content's structural integrity, such as edges and patch interrelations, independent of stylistic changes, rendering it an effective complement to perceptual metrics in style transfer evaluations.

## D  EXPERIMENT DETAILS

### D.1  DATASET

Our work utilizes two primary datasets: the MS-COCO 2017 Lin et al. (2015) dataset for content and the WikiArt Phillips & Mackintosh (2011) dataset for artistic styles. We use the 118,287 images from the MS-COCO 2017 training set as our content source, leveraging its rich variety of everyday scenes and objects. For our style library, we meticulously select 13 distinct artistic styles from WikiArt, which include: Oil painting, Kids' illustration, Watercolor, Ghibli, Landscape woodblock printing, Chinese Ink, Sketch, Pop art, Impressionism, Cubism, Cyberpunk, Pointillism, and Crayon.

To ensure experimental fairness across different style transfer expressions, we adopted specific strategies: 1) For exemplar-guided generation(*e.g.*, StyleID Chung et al. (2024)), we carefully selected paired images from the dataset, using style images as guidance; 2) For text-guided generation(*e.g.*, SMS Jiang et al. (2025)), we crafted appropriate prompts that accurately describe the same style images used in the exemplar approach, thereby facilitating high-quality generation. This dual approach allows for comprehensive evaluation of our method's versatility across different guidance modalities.

### D.2  IMPLEMENTATION DETAILS

We conduct experiments on a single NVIDIA RTX 4090 GPU with 24GB of VRAM. The software environment is built on Python 3.9, utilizing PyTorch 1.13.1 and CUDA 12.5 to leverage the GPU's computational power for accelerated processing. Crucially, our feature injection technique is applied starting from the 49th timestep. This strategic timing allows the model to first establish a strong content structure before introducing detailed style information, preventing the style from overwhelming the original content. Our method can be applied to various diffusion-based models; however, for fair comparison, we chose to use v1.4.

### D.3  OPTIMAL OBJECTIVES

#### D.3.1  GRAM MATRIX

In style transfer tasks, a central challenge lies in accurately capturing the style characteristics of an image, particularly global features such as texture and color. Traditional pixel-level operations fail to capture these global statistics and cannot disregard spatial information. To solve this problem, the Gram matrix Gatys et al. (2015) is introduced. The Gram matrix extracts statistical information by calculating the inner product between every pair of feature maps. Specifically, the Gram matrix is defined as:

$$G_{ij} = \langle f_i, f_j \rangle = \sum_k f_i(k) f_j(k), \tag{16}$$

effectively captures the second-order statistical information of feature vectors, encapsulating global distribution patterns such as texture and color while disregarding spatial positions.

The Gram matrix can be considered as a two-dimensional covariance matrix, capturing the correlations between different feature channels. In their seminal work, Gatys et al. demonstrated that two-dimensional covariance is particularly well-suited for style transfer tasks. They showed that

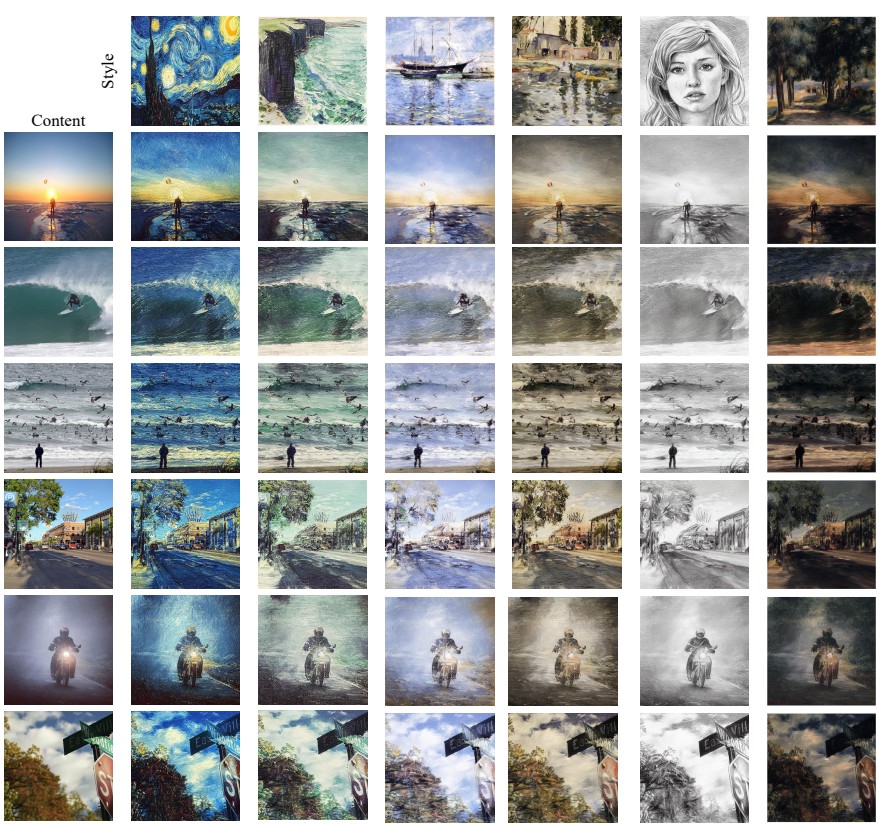

Figure 8: Visualization results of our proposed methods.

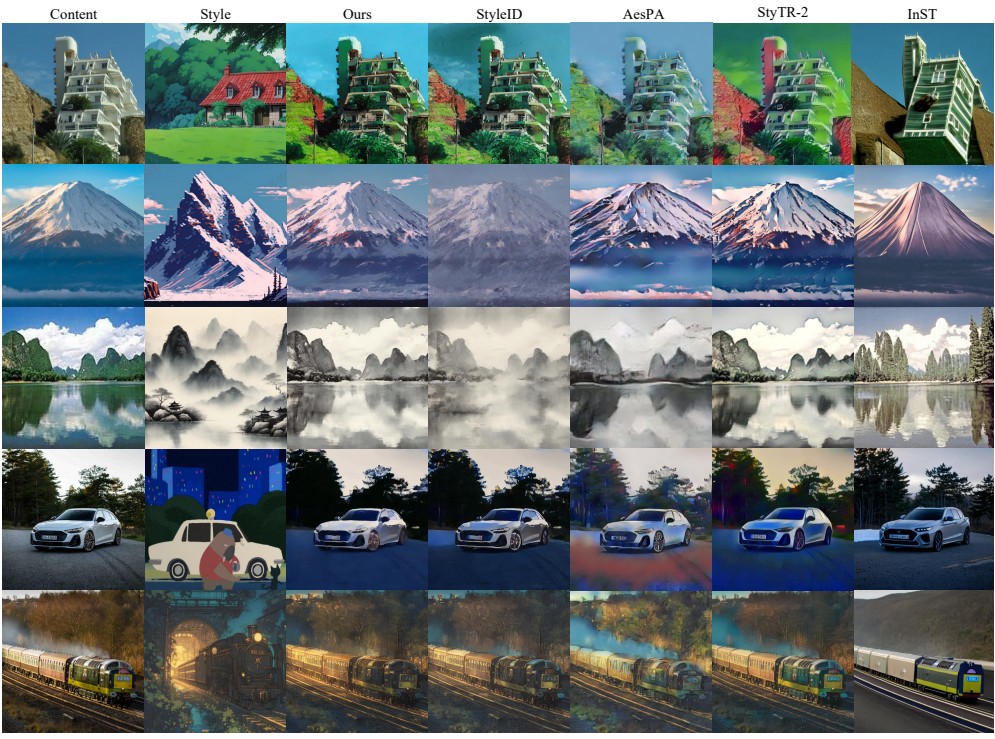

Figure 9: Comparison of style transfer results. Visual results of our CoCoDiff method compared against four baseline approaches across diverse artistic styles: Ghibli Style, Pixel Art, Chinese Ink, Kid Illustration, and Vintage Style.

among different types of covariance, two-dimensional covariance, as represented by the Gram matrix, is the most effective for describing style.

In style transfer applications, the Gram matrix facilitates optimization by minimizing distributional differences in feature spaces, enabling target images to emulate the distinctive patterns present in style reference images. The scale-invariance and robustness of Gram matrices establish them as ideal tools for style description, allowing for effective style transfer across diverse visual domains regardless of content structure or dimensional variations.

### D.3.2 SOBEL

A critical balance must be maintained in style transfer: on one hand, the algorithm must effectively learn and incorporate the distinctive feature information from the style reference image, while on the other hand, it must preserve the structural integrity of the content image without distortion. This dual objective presents a fundamental challenge in the field.

The Sobel operator, defined by convolving an image with kernels for vertical edges, effectively detects intensity gradients to highlight edges and line textures in generated images. By computing the gradient magnitude, typically as ( $\sqrt{G_x^2 + G_y^2}$ ), it emphasizes regions of rapid intensity change, which correspond to line structures and textures. This edge-enhancing capability allows the Sobel operator to control and refine the linear patterns and textural details in image generation, ensuring that stylistic elements like contours and boundaries are preserved or accentuated.

## E   MORE COMPARISONS WITH PATCHED-BASED METHODS

We include representative methods such as CNNMRF Li & Wand (2016), Style-Swap Chen & Schmidt (2016), etc. and conduct systematic quantitative comparisons. As shown in Tab. 6 and Fig. 6, CoCoDiff (Ours) achieves superior or highly competitive performance across all metrics. These results indicate that our method not only captures meaningful correspondences but also benefits from diffusion-based semantic alignment, enabling performance beyond conventional patch-level approaches. Importantly, our method is also training-free.

| Metric | Ours | CNNMRF | Style-Swap | DIA | Avatar-Net | SCSA+StyleID |
|--------|------|--------|-----------|------|-----------|-------------|
| FID ($\downarrow$) | 18.432 | 27.872 | 35.642 | 31.933 | 22.356 | 20.835 |
| LPIPS ($\downarrow$) | 0.549 | 0.672 | 0.793 | 0.661 | 0.641 | 0.562 |
| CFSD ($\downarrow$) | 0.609 | 0.844 | 0.761 | 0.649 | 0.753 | 0.612 |

Table 6: Comparison of metrics across methods.

## F   VISUAL COMPARISONS RESULTS

We present additional experimental results in Fig. 7 that showcase classic visual outcomes across a diverse range of artistic styles. Our method, CoCoDiff, effectively transfers styles from several artistic movements, including *Primitive Art, Romanticism, Impressionism, Post-Impressionism, Cartoon, and Sketch*. More results can be found in Fig. 8.

Further experiments confirm the robustness and versatility of our proposed method. The results demonstrate its ability to produce visually compelling stylized images while preserving high content fidelity. CoCoDiff maintains a superior balance between expressive styling and content preservation across a wide range of artistic scenarios.

Additionally, we present a comprehensive evaluation of our style transfer method (CoCoDiff) by comparing it with four established baseline approaches: StyleID Chung et al. (2024), AesPA-Net Hong et al. (2023), StyTR² Deng et al. (2022) and InST Zhang et al. (2023b). We test these methods across diverse artistic styles, including *Ghibli Style, Pixel Art, Chinese Ink, Kid Illustration, and Vintage Style* as shown in Fig. 9. Our experiments show that CoCoDiff consistently outperforms these baselines, achieving an outstanding balance of vivid stylistic rendering and accurate content preservation in each style. For example, in Ghibli Style, CoCoDiff captures the fluid, whimsical

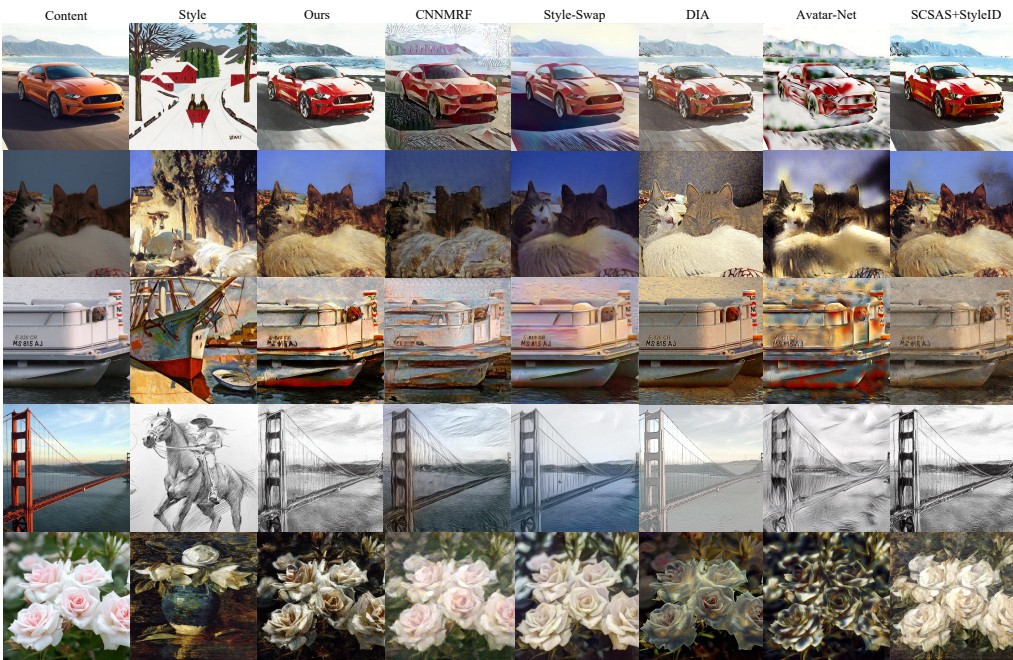

Figure 10: Additional comparisons with patch-based methods.

aesthetic more effectively than others, while in Chinese Ink, it preserves intricate brushstroke details with greater fidelity than StyleID. Similarly, for Pixel Art and Kid Illustration, CoCoDiff produces sharp, stylized images with minimal structural distortion compared to StyTR² and InST. These findings underscore CoCoDiff's adaptability and precision, ensuring visually striking and structurally coherent outputs across a broad spectrum of artistic domains.reinforcing its effectiveness across diverse artistic contexts.

### F.1 FINE-GRAINED FEATURE CORRESPONDENCE

A key contribution of our approach is the implementation of fine-grained feature correspondence. To clearly demonstrate this capability, we have included detailed style transfer results on three distinct subjects: flower, cow and house. These images in Fig. 11, Fig. 12 and Fig. 13 effectively illustrate how our method precisely aligns and transfers stylistic elements while meticulously preserving the unique content of each subject.

## G IMPACT

We believe our proposed training-free style transfer matching module achieves remarkable results through its novel approach. The concept of cyclic consistency can be readily integrated to various other style transfer methods to achieve high-fidelity feature correspondence. From a societal perspective, our work brings positive implications for entertainment devices, animation media, and related fields, while simultaneously raising important considerations regarding copyright protection and intellectual property rights.

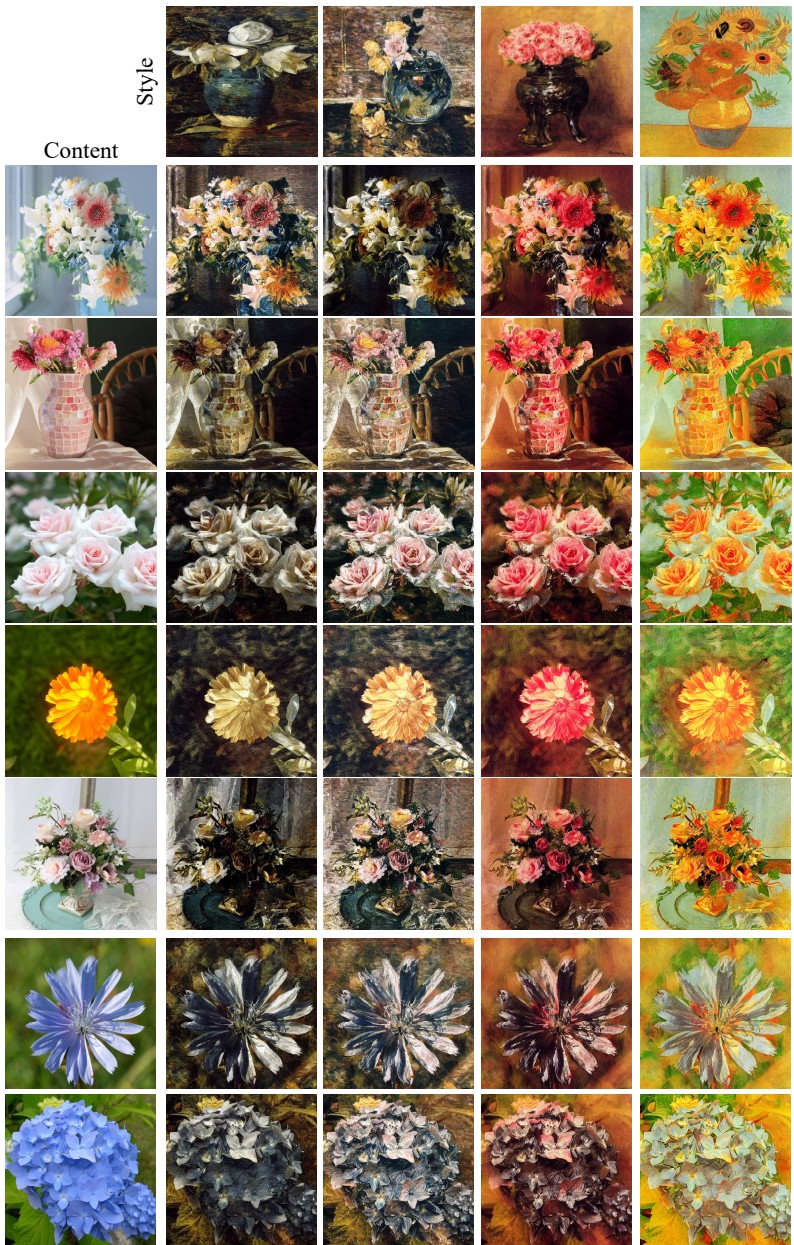

Figure 11: Fine-grained style transfer visual results: Flower.

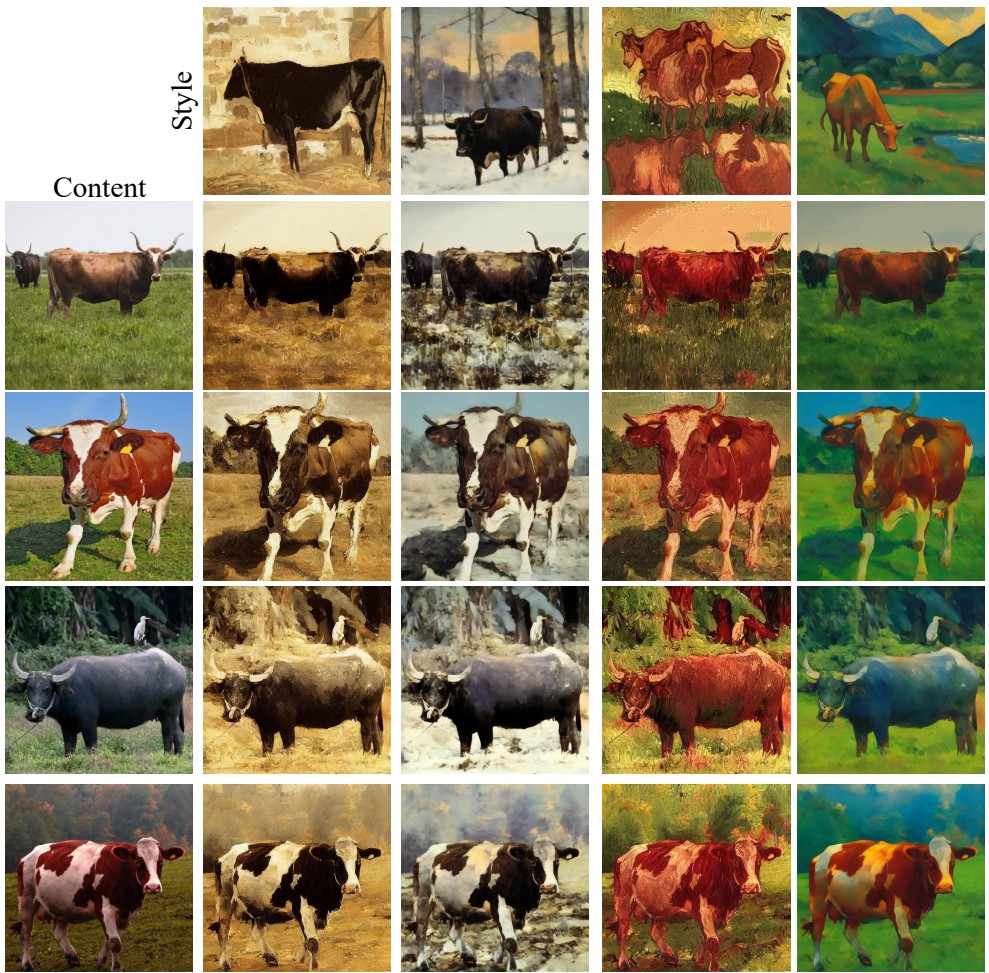

Figure 12: Fine-grained style transfer visual results: Cow.

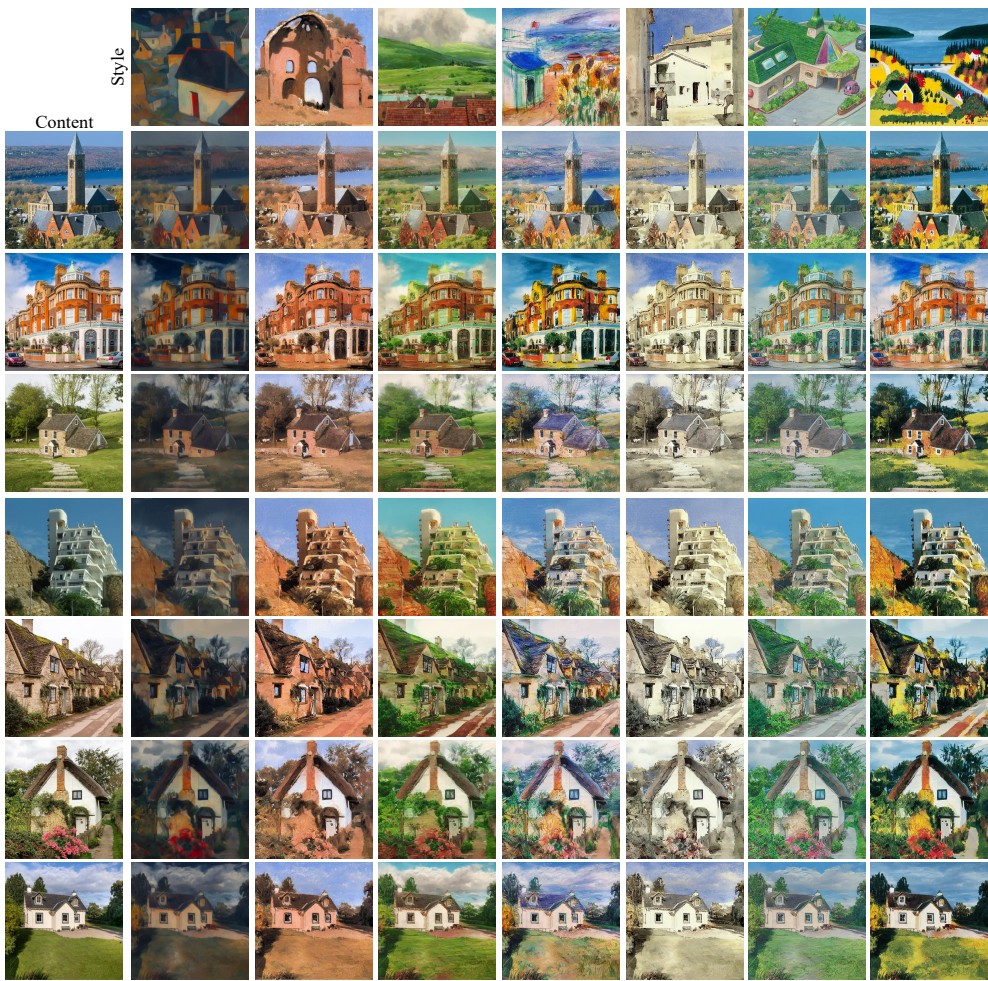

Figure 13: Fine-grained style transfer visual results: House.

