# OpenReview forum: "CoCoDiff: Correspondence-Consistent Diffusion Model for Fine-grained Style Transfer"
_ICLR.cc/2026/Conference — ICLR 2026 Poster_

### Official Review · Reviewer_1orb · 2025-10-15

**Soundness:** 2
**Presentation:** 3
**Contribution:** 2
**Rating:** 4
**Confidence:** 5

**Summary:**

This work proposed a training-free diffusion-based framework for fine-grained, correspondence-consistent style transfer. It leverages pretrained diffusion models to extract pixel-level semantic correspondences between content and style images, enabling region- and object-aware stylization.

The main contribution includes 1. Pixel-Wise Semantic Correspondence Module to extract intermediate diffusion features to build dense, semantically meaningful alignment maps between content and style images; and 2. Cycle-Consistency Optimization which
integrates attention-guided feature injection with iterative refinement to enhance structural stability and stylization coherence.

**Strengths:**

1. This work has enough metrics to prove that the solution is effective, across FID, LPIPS, ArtFID, and CFSD. The ablation and user studies further validate design choices such as the feature injection weight and AdaIN harmonization.

2. This paper is well written and easy to follow.

**Weaknesses:**

1. This work claimed training free but it still need iteration during sampling. the complexity is not stated in the work
2. Some innovations were introduced rather suddenly, lacking sufficient theoretical foundation and clear explanations.
3. Why sobel and g works for style and context similarity. I look forward to a more reasonable explanation
4. This work lacks many recent references. I hope the author can include it as a baseline for comparison. e.g. [1] [2]


[1] Ahn, Namhyuk, et al. "Dreamstyler: Paint by style inversion with text-to-image diffusion models." aaai24

[2] He, Huiang, et al. "Semantix: An Energy Guided Sampler for Semantic Style Transfer." iclr25

**Questions:**

1. I hope the author can provide the specific time and memory consumption of the method.
2. The work lacks ablation experiments. I hope the authors can demonstrate the role of each module they proposed. And provide more credible theoretical explanations for sobel and g network.
3. I hope the author can add some newer baselines. Or provide some objective theoretical analysis to explain the similarities and differences.

---

> ### Author Response · Authors · 2025-11-23
> **Our Detailed Response to Reviewer 1orb: Addressing Efficiency, Theoretical Basis, and Ablation Studies**
>
> ## To 1orb
> We are grateful for the thorough and constructive review provided by the reviewer. We address your concerns and questions below：
>
> ### W1 Q1: Time and Memory Consumption
> In response to the reviewer's concern regarding efficiency and practical applicability, we provide detailed information on both the inference time and memory consumption of our method. To ensure a fair comparison, all experiments are conducted on the same server and GPU configuration, utilizing the same dataset and default parameters across all approaches. The resulting average inference time comparison is presented in Table as below. Furthermore, the peak VRAM consumption for our proposed method is 23169 MiB.
>
> **Table 1: Runtime comparison with diffusion-based style transfer methods.**
>
> | Method | Ours | StyleID | InST | Instant+ | FreeStyle | SMS | DiffStyle | DiffuseIT |
> | :--- | :---: | :---: | :---: | :---: | :---: | :---: | :---: | :---: |
> | Time (s) | 24.6 | 10.8 | 18.5 | 39.9 | 27.5 | 205.9 | 320.7 | 786.8 |
>
> **Table 2: Runtime comparison with non-diffusion-based style transfer methods.**
>
> | Method | Ours | StyTR-2 | AesPA | AdaIN |
> | :--- | :---: | :---: | :---: | :---: |
> | Time (s) | 24.6 | 0.3 | 0.1 | 0.1 |
>
> ### W2: Theoretical Analysis
> We thank the reviewer for raising the concern about the theoretical foundation. To clarify, our method decomposes the intermediate diffusion features into a semantic component and a style-related residual:
>
> $$
> F_c^{t,l}(p_c) = u_c(p_c) + r_c(p_c), \quad F_s^{t,l}(p_s) = u_s(p_s) + r_s(p_s),
> $$
>
> where $(u_c, u_s)$ capture semantics and $(r_c, r_s)$ represent style-related variations. Direct cosine similarity between these features introduces cross-terms ($\langle u_c, r_s \rangle$, $\langle r_c, r_s \rangle$) that bias the matching towards texture rather than semantic alignment.
>
> To address this, our cycle consistency step applies attention-guided inverse style injection, which adjusts the style residual to be closer to the content residual, reducing style-related noise. This operation effectively creates a *semantic alignment domain*, where cosine similarity reflects semantic correspondence, not just texture similarity.
>
> Empirically, Fig. 4(B) shows that without cycle consistency, similarity maps are dominated by stylistic textures. After applying the cycle consistency step, the maps focus on true semantic regions, validating our theoretical analysis. This enhances the stability and locality of semantic correspondence, supporting the effectiveness of our approach.
>
> We also use a correspondence metric $M(t, l)$ to select the optimal feature space for each timestep $t$ and layer $l$, ensuring an appropriate balance between semantic consistency and stylistic fidelity. The final dense correspondence field is then used for the attention-based style transfer process.

---

> > ### Author Response · Authors · 2025-11-23
> >
> > ### W3 Q2: Ablation Experiments
> >
> > We design a set of targeted ablation experiments to verify each module’s individual contribution and their interactions. First, we compare four basic configurations (baseline, Sobel-only, Gram-only, and Sobel+Gram) to quantify Sobel’s role in structure preservation, Gram’s role in style promotion, and the trade-offs when they are combined. We evaluated all variants using FID, LPIPS, and CFSD to show how each module affects correspondence quality and structural fidelity. As reported in the table, the Gram-only configuration substantially reduces FID but degrades LPIPS, indicating that pure emphasis on style alignment comes at the cost of structural fidelity. By contrast, Sobel-only improves LPIPS and CFSD, suggesting that Sobel acts as an effective structural safeguard that limits excessive style injection. Combining Sobel and Gram yields a balanced compromise across all three metrics and achieves the best overall trade-off, supporting the conclusion that Sobel and Gram play complementary roles in driving style transfer while preserving structure.
> >
> > **Table 3: Results for Sobel and Gram combinations.**
> >
> > | Sobel | Gram | FID | LPIPS | CFSD |
> > | :---: | :---: | :---: | :---: | :---: |
> > | - | - | 26.513 | 0.697 | 0.804 |
> > | ✓ | - | 29.845 | 0.506 | 0.631 |
> > | - | ✓ | 23.471 | 0.753 | 0.761 |
> > | ✓ | ✓ | 18.432 | 0.549 | 0.609 |
> >
> >
> > ### W4 Q3: More Comparison
> > We compare our method with Dreamstyler [1] and Semantix [2]. As shown in the Tab. 4, CoCoDiff shows more stable performance in both overall generation quality and structural preservation.
> > It is important to note that Semantix [2] is not open-sourced. Although we attempt to reimplement it based on the details provided in the paper, the absence of released model weights and full implementation specifications prevents us from reproducing the results reported by the authors.
> >
> > **Table 4: Comparison of metrics across methods.**
> >
> > | Metric | CoCoDiff | Dreamstyler | Semantix |
> > | :--- | :---: | :---: | :---: |
> > | FID (↓) | 18.432 | 32.634 | -- |
> > | LPIPS (↓) | 0.549 | 0.793 | -- |
> > | CFSD (↓) | 0.609 | 0.754 | -- |
> >
> > [1] Ahn, Namhyuk, et al. "Dreamstyler: Paint by style inversion with text-to-image diffusion models." aaai24
> >
> > [2] He, Huiang, et al. "Semantix: An Energy Guided Sampler for Semantic Style Transfer." iclr25

---

> > > ### Comment · Reviewer_1orb · 2025-11-27
> > >
> > > I appreciate the author response. The reply addressed most of my concerns.
> > >
> > > Although I think the manuscript quality of this work still needs improvement, such as the writing logic, overall I would lean towards accepting it. Thus I raise my score to 6.

---

> > > > ### Author Response · Authors · 2025-11-27
> > > >
> > > > Thank you very much for your approval. Thanks again for your insightful comments and suggestions.

---

### Official Review · Reviewer_yGas · 2025-10-20

**Soundness:** 2
**Presentation:** 2
**Contribution:** 2
**Rating:** 2
**Confidence:** 4

**Summary:**

This paper focuses on enhancing the performance of artistic style transfer between images with similar semantics. A key limitation of most existing works is that they operate at the global level, while overlooking region-wise and even pixel-wise semantic correspondence. To address this gap, the authors propose CoCoDiff, a training-free diffusion-based framework. This framework extracts intermediate diffusion features to establish pixel-wise correspondences and leverages cyclic optimization techniques, achieving fine-grained stylization with semantic consistency.

**Strengths:**

+ The integration of semantic consistency into diffusion-based style transfer is interesting and beneficial for stylization tasks between images with similar semantics.

+ The proposed method is training-free.

**Weaknesses:**

- My primary concern is that this paper appears to overlook a crucial research direction in style transfer, namely patch-based style transfer. Aligned with the motivation of this paper, works in this direction primarily leverage semantic correspondence between features to perform style transfer between objects with similar semantics. Representative methods include CNNMRF [A], Style-Swap [B], DIA [C], DivSwapper [D], Avatar-Net [E], and SCSA [F]. This paper neither discusses these methods in the related work section nor compares with them in experiments. Consequently, it is hard to effectively evaluate the technical innovation and performance of this work.

[A] Chuan Li and Michael Wand. Combining markov random fields and convolutional neural networks for image synthesis. In CVPR, pages 2479–2486, 2016.

[B] Tian Qi Chen and Mark Schmidt. Fast patch-based style transfer of arbitrary style. arXiv preprint
arXiv:1612.04337, 2016.

[C] Jing Liao, Yuan Yao, Lu Yuan, Gang Hua, and Sing Bing Kang. Visual attribute transfer through deep image analogy. TOG, 2017.

[D] Zhizhong Wang, Lei Zhao, Haibo Chen, Zhiwen Zuo, Ailin Li, Wei Xing, and Dongming Lu. Divswapper: Towards diversified patch-based arbitrary style transfer. In IJCAI, pages 4980–4987, 2022.

[E] Lu Sheng, Ziyi Lin, Jing Shao, and Xiaogang Wang. Avatarnet: Multi-scale zero-shot style transfer by feature decoration. In CVPR, pages 8242–8250, 2018.

[F] Chunnan Shang, Zhizhong Wang, Hongwei Wang, Xiangming Meng. SCSA: A Plug-and-Play Semantic Continuous-Sparse Attention for Arbitrary Semantic Style Transfer. In CVPR, pages 13051–13060, 2025.

- In L247–248, the authors state: “The optimal pair $(t^∗, l^∗)$ is selected by maximizing a correspondence quality metric $\mathcal{M}(t, l)$ over predefined candidate sets $\mathcal{T}$ and $\mathcal{L}$” How are $\mathcal{T}$ and $\mathcal{L}$ determined? Are they set based on empirical values?

- It seems that the reason that the proposed method can perform style transfer in a training-free manner mainly relies on the adjustment of attention weights in Eq. (7). Why can this adjustment effectively enhance the semantic consistency of the feature maps? Some necessary theoretical explanations are lacking.

- In L314–316, it is mentioned that “the fitting cycle's iteration process terminates when both ... are predefined thresholds.” It is unclear what kind of constraint a content loss ($\mathcal{L}_{content}$) greater than a threshold ($\tau_c$) can impose. What is the difference between this and constraining only the style loss?

- Section 4.6 (User Study) lacks necessary details. For example: what were the scoring instructions provided to users? What were the scoring rules? Which specific aspects were evaluated under the “Style” and “Content” dimensions in Table 3?

- Some minor detail issues: In L144–145, ArtFlow does not belong to the diffusion-based methods. In Eq. (6), $F_s$ is not defined.

**Questions:**

Please see weaknesses above.

---

> ### Author Response · Authors · 2025-11-23
> **Our Detailed Response to Reviewer yGas: Comparative Analysis and Design Rationale Validation**
>
> ## To yGas
> We would like to thank the reviewer for involving in the discussion and providing us with some important feedback. Below, we further address your concerns:
>
> ### W1: More Comparison.
> We include representative methods such as CNNMRF, Style-Swap, etc. and conduct systematic quantitative comparisons. As shown in Tab. 1, CoCoDiff (Ours) achieves superior or highly competitive performance across all metrics. These results indicate that our method not only captures meaningful correspondences but also benefits from diffusion-based semantic alignment, enabling performance beyond conventional patch-level approaches. Importantly, our method is also training-free.
> We respectfully note that the DivSwapper [D] does not release its complete evaluation code which makes certain metrics unavailable. We include the others to ensure that the comparison is as comprehensive and fair as possible.
>
> **Table 1: Comparison of metrics across methods.**
>
> | Metric | CoCoDiff (Ours) | CNNMRF | Style-Swap | DIA | Div | Avatar-Net | SCSA+StyleID |
> | :--- | :---: | :---: | :---: | :---: | :---: | :---: | :---: |
> | FID (↓) | 18.432 | 27.872 | 35.642 | 31.933 | -- | 22.356 | 20.835 |
> | LPIPS (↓) | 0.549 | 0.672 | 0.793 | 0.661 | -- | 0.641 | 0.562 |
> | CFSD (↓) | 0.609 | 0.844 | 0.761 | 0.649 | -- | 0.753 | 0.612 |
>
> ### W2: Correspondence Quality $M$
> Empirically, larger timesteps $t$ and earlier U-Net layers $l$ typically capture more semantic features, while smaller timesteps and deeper layers contain more low-level information. Style transfer requires an appropriate balance between the semantic consistency of the content image and the stylistic fidelity of the reference image. To achieve this, we select the optimal feature space by evaluating each $(t, l)$ pair using the correspondence quality metric $M(t, l)$, which measures the accuracy of cross-image feature matching. We evaluate our method on the popular benchmark datasets: **SPair-71k** [1], and use percentage of correct keypoints (PCK) as the evaluation metric to assess the accuracy of semantic correspondences. This allows us to identify the optimal values for $t$ and $l$.
>
> These selections are based on an intuitive understanding of the diffusion process and are further validated through experimental results. Thus, we believe that this empirical approach significantly enhances the performance of style transfer.
>
> ### W3: Attention weights explanations
> A key point we would like to clarify is that adjusting attention weights to influence editing behavior has long been a common practice in training-free diffusion-based image editing. Prompt-to-Prompt [2] was the first to explicitly demonstrate that modifying cross-attention weights can steer editing results. Subsequent methods such as MasaCtrl [3], StyleID [4], and many other frameworks, have continued to follow this paradigm.
>
> However, uniformly modifying all attention maps inevitably alters regions that should remain unchanged, which precisely highlights the necessity of our approach. By explicitly establishing correspondence between source and target structures, our method adjusts attention only along the aligned correspondence-aware attention paths. This targeted modulation is more principled and leads to significantly better editing fidelity.
> To address this concern, we conduct an ablation study that isolates the contribution of the key (K) and value (V) branches in the attention-guided feature injection. The results in Tab. 2 show that replacing K only breaks semantic alignment, leading to severe degradation of structure. Replacing V only strengthens stylization but introduces noticeable structural distortion . Only the joint replacement of K and V achieves the best performance across all metrics (FID 18.432, LPIPS 0.549), preserving both stylistic coherence and structural stability.
>
> **Table 2: Effect of key/value injection on semantic alignment.**
>
> | K | V | FID | LPIPS |
> | :---: | :---: | :---: | :---: |
> | ✓ | - | 47.500 | 0.802 |
> | - | ✓ | 33.986 | 0.854 |
> | ✓ | ✓ | 18.432 | 0.549 |

---

> > ### Author Response · Authors · 2025-11-23
> >
> > ### W4: Content Loss and Style Loss
> > We clarify that the content loss serves to prevent excessive degradation of content structure during the fitting cycle. In our iterative optimization, continuously reducing the style loss pushes the output toward stronger stylization, but this also tends to weaken the semantic structure of the content image. When the content loss exceeds a certain upper bound, it indicates that the process is moving into a stage where the structure begins to deteriorate. Therefore, the content-loss functions as a structural safeguard that stops the iteration before the content becomes overly distorted.
> > This is fundamentally different from constraining only the style loss. This complementary effect is also confirmed in our ablation study. As shown in Tab. 3, using only content-aware constraint or only style-aware constraint does not yield the best performance. These findings demonstrate that the dual constraints on style and content jointly prevent both under-stylization and excessive stylization, which is precisely why both thresholds are necessary in the fitting cycle.
> > We will clarify this mechanism more explicitly in the revised manuscript.
> >
> > **Table 3: Results for Sobel and Gram combinations.**
> >
> > | Sobel | Gram | FID | LPIPS | CFSD |
> > | :---: | :---: | :---: | :---: | :---: |
> > | - | - | 26.513 | 0.697 | 0.804 |
> > | ✓ | - | 29.845 | 0.506 | 0.631 |
> > | - | ✓ | 23.471 | 0.753 | 0.761 |
> > | ✓ | ✓ | 18.432 | 0.549 | 0.609 |
> >
> > ### W5: User Study
> > This study invites 25 non-expert participants (aged 19–45). Each participant views multiple sets of content and style images, together with the stylized results generated by different methods. To avoid bias, the names of all methods are hidden, the display order is fully randomized, and all images are presented at a uniform resolution. Participants rank all images in each set based on two explicit evaluation criteria: “Style Fidelity,” which measures how faithfully the stylized image reflects the color tone, texture strokes, and overall visual style of the reference style image; and “Content Preservation,” which measures whether the main structures, edges, and semantic regions are well preserved during the transfer, making it clear that the image undergoes style transfer rather than being redrawn. For each set of images, participants perform preference ranking according to these two criteria, allowing direct comparison among different methods and reducing bias caused by individual differences in scoring. Finally, the ranking results of all methods along the two dimensions are aggregated, and the outcomes are presented in Tab. 3.
> >
> > ### W6: Minor details
> > We recognize the issues and have corrected them in the revised manuscript.
> >
> > [1]J. Min et al., SPair-71k: A Large-scale Benchmark for Semantic Correspondence.   ICCV 2019.
> >
> > [2]A. Hertz et al., Prompt-to-Prompt Image Editing with Cross Attention Control. arXiv preprint arXiv:2208.01626, 2022.
> >
> > [3]M. Cao et al., MasaCtrl: Tuning-Free Mutual Self-Attention Control for Consistent Image Synthesis and Editing. ICCV 2023.
> >
> > [4]Jiwoo Chung et al., Style injection in diffusion: A training-free approach for adapting large-scale diffusion models for style transfer. CVPR 2024.

---

> > > ### Comment · Reviewer_yGas · 2025-11-26
> > > **Post-rebuttal comment**
> > >
> > > I appreciate the authors’ efforts in the rebuttal. Most of my concerns have been addressed. However, my major concern lingers regarding the discussions and comparisons with the patch-based style transfer research line. The authors only show quantitative results in the rebuttal, and this research direction is still overlooked in the revised manuscript, absent from both the related work section and the qualitative/quantitative comparison results.

---

> > > > ### Author Response · Authors · 2025-11-27
> > > >
> > > > Thank you for your thoughtful feedback and for acknowledging the improvements in our rebuttal. We appreciate your continued engagement with our work.
> > > >
> > > > Regarding your concern about the patch-based style transfer research line, we would like to inform you that we have addressed this in the revised manuscript. Specifically, we have updated the Related Work section to include a more comprehensive discussion of patch-based style transfer approaches. Additionally, we have incorporated both qualitative and quantitative comparisons with this research line in the main body of the manuscript as well as in the appendix.
> > > >
> > > > We hope these additions adequately address your concern and provide a more complete comparison with existing approaches. Thank you again for your valuable feedback.

---

### Official Review · Reviewer_gHaT · 2025-10-29

**Soundness:** 2
**Presentation:** 2
**Contribution:** 2
**Rating:** 4
**Confidence:** 4

**Summary:**

This paper  proposes a training-free, diffusion-based framework for fine-grained, structure-preserving style transfer that operates directly on pretrained backbones without additional supervision or fine-tuning. The propose method delivers state-of-the-art visual quality and strong quantitative results, outperforming methods that rely on extra training or annotations.

**Strengths:**

(1) The proposed training-free style transfer method, which ensures style consistency, represents a valuable contribution to the field.
(2) A clear algorithm is presented for better understanding.
(3) CoCoDiff outperforms the six representative methods to some extend.

**Weaknesses:**

(1) Line 252 says M(t, l) that evaluates the alignment quality based on the extracted feature maps at timestep t and layer l. What exactly is M(t, l)?

(2) Fig. 2 requires significant refinement for improved clarity. FFM appears to have two inputs, but only I_sty_c is explicitly shown in the main framework. It seems that I_sty is inputted and reconstructed within the U-Net. If this is used for self-attention extraction, I suggest that the output should also be included. If this is the case, the self-attention section should be moved from Section 3.1 to Section 3.2 for better understanding. Besides, FIC also has two inputs where I can see only one.

The structure of Fig.2 should be better organized. The input images are put in the center, which is not easy to understand

(3) The used SDv1.4 is a very old version. Do the authors try newer diffusion models like SD2.1, SDXL or FLUX.

(4) The inference time should be listed as it consists of many complex steps like feature exchange, fitting cycle, iterative control.

**Questions:**

Please refer to the weakness.

**Details Of Ethics Concerns:**

I have no specific concerns.

---

> ### Author Response · Authors · 2025-11-23
> **Our Response to Reviewer gHaT: Detailed Experimental Results and Clarifications on Core Components**
>
> ## To gHaT
> We sincerely thank the reviewer for the insightful question. We address your concerns and questions below:
>
> ### W1: Correspondence Quality $M$.
> Here, we provide a clear explanation of the definition and quantification of $M(t,l)$. To determine whether the diffusion features extracted at timestep $t$ and U-Net layer $l$ are suitable for constructing reliable cross-image correspondences, we introduce the correspondence quality metric $M(t,l)$. This metric evaluates the semantic alignment accuracy when performing cross-image feature matching using cosine similarity within the feature space defined by $(t,l)$.
>
> Specifically, we follow the standard semantic correspondence protocol and extract diffusion features from image pairs on benchmark datasets such as SPair-71k [1]. For each content keypoint, we identify the location in the style image that achieves the highest cosine similarity and use **Percentage of Correct Keypoints (PCK)** as the evaluation criterion to determine whether the predicted keypoint falls within an acceptable distance from the ground-truth correspondence. The metric $M(t,l)$ is defined as the average PCK score over all evaluated samples, reflecting the reliability of semantic correspondence within the feature space $(t,l)$.
>
> We then perform a grid search over the candidate sets $\mathcal{T}$ and $\mathcal{L}$ and select the combination that maximizes $M(t,l)$ as the final choice of $(t^\*, I^\*)$ . Therefore, $M(t,l)$ serves as a direct quantitative indicator of "correspondence quality," where higher values indicate a feature space that is more suitable as the semantic alignment foundation for subsequent style transfer.
>
> ### W2: Figure 2.
> We further clarify that the FFM module has two inputs $I_{sty}$ and $I_{sty_c}$. In the U-Net branch, $I_{sty}$ is not altered in structure. It is only used to extract self-attention maps. For FIC, intermediate result at each iteration is the only one direct input. During each iteration, this intermediate result is compared with the fixed reference images $I_{sty}$ and $I_{con}$ to compute the required constraints.
>
> We will revise Fig. 2 to provide a clearer illustration and eliminate any possible confusion.

---

> ### Author Response · Authors · 2025-11-23
>
> ### W3: Different Diffusion Models.
> We conduct preliminary experiments with updated backbones including SD v1.5, SD v2.1, and SDXL, as shown in the table below. The results show that CoCoDiff exhibits strong flexibility and generalizability across different backbone models, mainly due to the stronger generative priors and richer semantic representations provided by these newer models.
>
> However, in the main results reported in the paper, we intentionally choose SD v1.4. This is because most existing style transfer and diffusion-based baselines (e.g., StyleID [2], InST [3]) are implemented and evaluated using SD v1.4. Using substantially stronger backbones—such as SDXL or FLUX—results in an unfair and less informative comparison, as the performance improvements may come from the backbone itself rather than from the CoCoDiff framework.
>
> In addition, due to architectural differences across models and limited time, we do not fully optimize larger models such as SDXL to reach their best performance. We believe that with more extensive experiments and hyperparameter tuning (e.g., gamma, temperature), we can identify more suitable configurations for these newer backbones and further improve overall performance.
>
> **Table: Comparison across different diffusion models.**
>
> | Metric | SD v1.4 | SD v1.5 | SD v2.1 | SD XL |
> | :--- | :---: | :---: | :---: | :---: |
> | FID (↓) | 18.432 | 17.995 | 18.252 | 21.412 |
> | LPIPS (↓) | 0.549 | 0.525 | 0.621 | 0.635 |
> | CFSD (↓) | 0.609 | 0.589 | 0.766 | 0.714 |
>
> ### W4: Inference time.
> We are pleased to provide the inference time comparison, which showcases the efficiency of our proposed method.The key to this advantage is that our design ensures zero training overhead, as it completely bypasses the need to train modules such as LoRA [4] or ControlNet [5].
>
> **Table: Runtime comparison with diffusion-based style transfer methods.**
>
> | Method | Ours | StyleID | InST | Instant+ | FreeStyle | SMS | DiffStyle | DiffuseIT |
> | :--- | :---: | :---: | :---: | :---: | :---: | :---: | :---: | :---: |
> | Time (s) | 24.6 | 10.8 | 18.5 | 39.9 | 27.5 | 205.9 | 320.7 | 786.8 |
>
> **Table: Runtime comparison with non-diffusion-based style transfer methods.**
>
> | Method | Ours | StyTR-2 | AesPA | AdaIN |
> | :--- | :---: | :---: | :---: | :---: |
> | Time (s) | 24.6 | 0.3 | 0.1 | 0.1 |
>
> [1]J. Min et al., SPair-71k: A Large-scale Benchmark for Semantic Correspondence. in Proc. ICCV 2019.
>
> [2]J. Chung et al., Style injection in diffusion: A training-free approach for adapting large-scale diffusion models for style transfer. CVPR 2024.
>
> [3]Y. Zhang et al., Inversion-Based Style Transfer With Diffusion Models. CVPR 2023.
>
> [4]E. J. Hu et al., LoRA: Low-Rank Adaptation of Large Language Models. ICLR 2022.
>
> [5]L. Zhang et al., Adding Conditional Control to Text-to-Image Diffusion Models. ICCV 2023.

---

### Official Review · Reviewer_hz3B · 2025-11-01

**Soundness:** 3
**Presentation:** 2
**Contribution:** 2
**Rating:** 6
**Confidence:** 3

**Summary:**

This paper proposes CoCoDiff, a training-free fine-grained style transfer framework based on pretrained latent diffusion models. The method establishes dense pixel-level semantic correspondences by mining intermediate diffusion features and introduces a cycle-consistency module to enforce structural and perceptual alignment. The approach achieves good visual quality and quantitative results.

**Strengths:**

1. The method demonstrates certain innovation by mining intermediate diffusion features to construct dense pixel-level semantic correspondences between content and style images, and introducing a cycle-consistency module to enhance structural and perceptual alignment.

2. The paper achieves superior quantitative and qualitative results compared to existing methods across multiple datasets.

**Weaknesses:**

1. Why does cycle consistency improve semantic correspondence? The paper lacks theoretical analysis or deeper explanation.

2. The correspondence quality metric M(t,l) in Equation (5) is not clearly defined. How is "correspondence quality" quantified?

3. Inconsistent notation usage: symbols such as p_c, p_s, p*_s are defined inconsistently across different sections.

4. Figure 8 has low comparison quality with text labels that are too small.

4. Insufficient experiments. Table 2 shows that on Mip-NeRF 360, GENIE's PSNR is 5-7 dB lower than Mip-NeRF, with even larger gaps in SSIM and LPIPS metrics. The authors attribute this to Gaussian volume density, but lack systematic analysis. The authors need to strengthen the persuasiveness.

5. Missing training time comparisons.

**Questions:**

1. The grid search for finding optimal (t*, l*) lacks theoretical guidance.

2. Why does "first converting the style image to content style" improve matching? Please provide theoretical analysis or more detailed mechanism explanation.

3. The authors need to provide sufficient validation on different diffusion models.

4. The authors need to report runtime and computational resource consumption.

5. How are the candidate sets T and L for grid search selected?

---

> ### Author Response · Authors · 2025-11-23
> **Our Detailed Response to Reviewer hz3B: Clarifications on Theory, Correspondence Metric, and Generalizability**
>
> ## To hz3B
> We thank the reviewer for the positive assessment and constructive suggestions. Please see the detailed response below.
>
> ### W1 Q2: Theoretical Analysis
> We thank reviewer for the positive assessment and constructive suggestions. In our framework, the intermediate diffusion features at timestep $t$ and layer $l$ can be decomposed into a semantic component and a style-related residual:
> $$
> F_c^{t,l}(p_c) = u_c(p_c) + r_c(p_c), \quad F_s^{t,l}(p_s) = u_s(p_s) + r_s(p_s),
> $$
> where $(u_c, u_s)$ encode object- and region-level semantics, while $(r_c, r_s)$ capture appearance statistics such as texture, brush strokes, and color. Consequently, directly computing cosine similarity,
> $$
> \cos(F_c^{t,l}(p_c), F_s^{t,l}(p_s)) = \frac{\langle F_c^{t,l}(p_c), F_s^{t,l}(p_s) \rangle}{\|F_c^{t,l}(p_c)\|\|F_s^{t,l}(p_s)\|},
> $$
> introduces several cross-terms such as $\langle u_c, r_s \rangle$ and $\langle r_c, r_s \rangle$, which dominate the similarity due to the large style-induced variance and bias the matching toward texture-level resemblance rather than semantic correspondence.
>
> Our cycle consistency step applies an attention-guided inverse style injection that maps the style image to an intermediate image($I_{sty_c}$) , whose feature representation becomes:
>
> $$
> F_{sty_c}^{t,l}(p) = u_s(p) + \tilde{r}_s(p),
> $$
>
> where the modified residual $\tilde{r}_s$ is statistically closer to the content residual. From a distributional viewpoint, this operation contracts the domain gap between content and style:
>
> $$
> D_{\mathrm{KL}}(P_c \| P_{sty_c}) < D_{\mathrm{KL}}(P_c \| P_s).
> $$
>
> Under this condition, the expected interference from style-related noise decreases,
>
> $$
> \mathbb{E}[|\langle u_c, \tilde{r}_s \rangle|] < \mathbb{E}[|\langle u_c, r_s \rangle|],
> $$
>
> making the semantic inner product $\langle u_c, u_s \rangle$ the dominant contributor during nearest-neighbor matching. In essence, cycle consistency constructs a *semantic alignment domain*, where cosine similarity more faithfully reflects semantic relatedness rather than appearance similarity.
>
> Importantly, this theoretical explanation is strongly supported by our empirical heatmap results as shown in Fig. 4(B). Without cycle consistency, the similarity maps exhibit large false-positive regions driven by stylistic textures. After applying the cycle step, these maps concentrate sharply on true semantic regions, and their spatial gradients become smoother and more coherent. The transition of high-response areas from texture-driven to structure-driven regions visually corroborates our mathematical analysis: cycle consistency effectively suppresses style-induced noise and enhances the stability and locality of semantic correspondence.
>
> ### W2 Q1 Q5: Correspondence Quality ($M$)
> Before establishing pixel-wise correspondences between the content and style images, it is essential to determine an appropriate feature space within the diffusion model.
> Empirically, larger timesteps $t$ and earlier U–Net layers $l$ tend to capture semantic features, whereas smaller timesteps and deeper layers have more low-level information. Since style transfer requires an appropriate balance between semantic consistency of the content and stylistic fidelity of the reference image, we determine the optimal feature space by evaluating each $(t,l)$ pair using a correspondence ($M(t,l)$), which measures the accuracy of cross-image feature matching. The form of this metric follows the correspondence evaluation protocol introduced in DIFT [1].
> We evaluate a small subset of samples from the dataset to select the optimal feature configuration as
>
> $$
> (t^\*, l^\*)=\arg\max_{t\in T, l\in L} M(t,l).
> $$
>
> After fixing the optimal timestep and layer, we extract the corresponding feature maps of both images and construct dense semantic correspondences. For every spatial position $p_c$ in the content feature map, we compute the cosine similarity between its feature vector and all spatial locations $p_s$ in the style feature map. The best match is defined as the position achieving the highest similarity, namely
> $$
> p_s^\*=\arg\max_{p_s \in I_s}\cos\big(p_c, p_s\big).
> $$
> This process yields a dense and semantically coherent correspondence map, which forms the basis for subsequent attention-based feature injection and fine-grained style transfer.

---

> > ### Author Response · Authors · 2025-11-23
> >
> > ### W4 W5 W6:
> > We provide clear explanations and supporting data for the questions raised by the reviewer. However, we politely note that some of the detailed issues addressed in the provided feedback do not appear to stem from the original set of questions in our paper's review. We ensure that our response comprehensively addresses the core technical aspects and design choices relevant to our submission.
> > ### Q3: Different Diffusion Models.
> > We further clarify the validation across different diffusion models as follows. To address this, we conduct preliminary experiments using several recent diffusion backbones, including SD v1.5, SD v2.1, and SDXL, as shown in the table below. These experiments show that CoCoDiff exhibits strong generalizability across various models, benefiting from the enhanced generative priors and richer semantic representations of these newer backbones.
> >
> > However, for the primary results presented in the paper, we intentionally choose SD v1.4 for the following reasons. First, most of the existing style transfer and diffusion-based baselines (such as StyleID [2], InST [3]) are implemented and evaluated using SD v1.4. Using substantially stronger backbones, such as SDXL, introduces an unfair comparison, as performance gains can stem from the backbone’s enhanced capabilities rather than from the CoCoDiff framework itself. For a more direct and comparable evaluation, we maintain consistency with these baselines by choosing SD v1.4.
> >
> > Furthermore, we acknowledge that more extensive experiments with newer diffusion models are necessary to fully explore their potential with CoCoDiff. While architectural differences and time constraints prevent us from fully optimizing these models, we believe that by adjusting hyperparameters such as gamma and temperature, we can further improve performance. These adjustments do not require retraining but can help identify more suitable configurations for these newer backbones. We plan to expand this analysis in future work, exploring different configurations and further validating CoCoDiff’s performance with these newer diffusion backbones.
> >
> > **Table 1: Comparison across different diffusion models.**
> >
> > | Metric | SD v1.4 | SD v1.5 | SD v2.1 | SD XL |
> > | :--- | :---: | :---: | :---: | :---: |
> > | FID (↓) | 18.432 | 17.995 | 18.252 | 21.412 |
> > | LPIPS (↓) | 0.549 | 0.525 | 0.621 | 0.635 |
> > | CFSD (↓) | 0.609 | 0.589 | 0.766 | 0.714 |
> >
> > ### Q4: Time and Computational Costs.
> > We compare our method with a set of baseline methods using the average generation time per image (seconds) in Tab. 2 and Tab. 3. The results show that our method achieves strong efficiency. It also does not require extra training or additional modules (e.g., LoRA [4], ControlNet [5]). Furthermore, CoCoDiff can benefit from existing diffusion-acceleration methods to achieve even faster inference. Our method uses 23169 MiB of GPU memory during inference.
> >
> > **Table 2: Runtime comparison with diffusion-based style transfer methods.**
> >
> > | Method | Ours | StyleID | InST | Instant+ | FreeStyle | SMS | DiffStyle | DiffuseIT |
> > | :--- | :---: | :---: | :---: | :---: | :---: | :---: | :---: | :---: |
> > | Time (s) | 24.6 | 10.8 | 18.5 | 39.9 | 27.5 | 205.9 | 320.7 | 786.8 |
> >
> > **Table 3: Runtime comparison with non-diffusion-based style transfer methods.**
> >
> > | Method | Ours | StyTR-2 | AesPA | AdaIN |
> > | :--- | :---: | :---: | :---: | :---: |
> > | Time (s) | 24.6 | 0.3 | 0.1 | 0.1 |
> >
> > [1]L. Tang et al., Emergent Correspondence from Image Diffusion. NIPS 2023.
> >
> > [2]J. Chung et al., Style injection in diffusion: A training-free approach for adapting large-scale diffusion models for style transfer. CVPR 2024.
> >
> > [3]Y. Zhang et al., Inversion-Based Style Transfer With Diffusion Models. CVPR 2023.
> >
> > [4]E. J. Hu et al., LoRA: Low-Rank Adaptation of Large Language Models. ICLR 2022.
> >
> > [5]L. Zhang et al., Adding Conditional Control to Text-to-Image Diffusion Models. ICCV 2023.

---

### Author Response · Authors · 2025-11-23
**Overall Response to Reviewer Feedback: Addressing Concerns and Summarizing Major Improvements**

## **General Response**

We thank all reviewers for their valuable and constructive feedback. We are encouraged that reviewers acknowledged the novelty of exploiting **intermediate diffusion features for dense pixel-level correspondence** (hz3B, gHaT, yGas), the **training-free nature** of our framework (yGas) , and the **strong qualitative and quantitative results** across diverse datasets(hz3B, gHaT, 1orb). Below we summarize the major concerns and outline the improvements we have made.

---

## **Method Novelty and Contribution**

We are pleased that reviewers recognized our key contribution:

1. Establishing **fine-grained semantic correspondence** by mining intermediate diffusion features.

2. Introducing a **cycle-consistency optimization** to enforce structural and perceptual alignment.

3. Enabling **training-free** fine-grained stylization that requires **no additional supervision** or model updates.


Reviewers also noted that the method is conceptually interesting and practically useful, especially for tasks requiring semantic preservation.

---

## **Main Improvements Made**

1.To address reviewer concerns, we clarified the theoretical intuition behind **cycle consistency**, expanded the explanation of the **correspondence quality metric M(t, l)**, and unified notation throughout the manuscript.

2.We added **newer diffusion backbones (SD1.5, SD2.1, SDXL)** and **ablation study for Sobel and Gram combinations** in the manuscript.

3.We enriched our discussion and comparisons with **patch-based semantic style transfer methods** (CNNMRF, StyleSwap, DIA, Avatar-Net, SCSA), and clarified the conceptual differences between our diffusion-based dense correspondence mechanism and patch-sampling methods.

4.We further strengthened the empirical evaluation by adding runtime and memory profiling. These updates show the effectiveness and efficiency of CoCoDiff relative to training-based and training-free baselines.

## **Other Additions to the Paper**

Additional improvements in the revised paper include:

1. A explanation of the correspondence quality metric M(t, l)

2. Update and unified notation throughout the manuscript

3. Added newer diffusion backbones (SD2.1, SDXL) in Tab.2 and visual results in Appendix

4. Added ablation study for Sobel and Gram combinations in Tab.3

---

### Author Response · Authors · 2025-12-02
**Summary of our current rebuttal status**

Dear Reviewers, ACs, SACs, and PCs,

We sincerely appreciate the reviewers’ thoughtful feedback and their active engagement throughout the rebuttal and discussion period. Throughout the period from Nov. 19 to Nov. 27, 2025, we responded to the reviewers’ questions with additional explanations, extended analyses, and further experiments in order to strengthen and clarify the contributions of our work.

Before the scoring reset caused by the system issue, the reviewers had already reacted to our clarifications. Reviewer 1orb had raised the score from 4 to 6. Reviewer yGas stated that most of the earlier questions had been satisfactorily addressed. The remaining two reviewers had not yet responded to our rebuttals. All of these developments, as well as our final messages prior to the system error.

Below, we briefly summarize what we clarified and added for each reviewer during the discussion period:

### **Reviewer 1orb (score updated from 4 to 6, last responded before the bug)**

- Reviewer 1orb explicitly stated that the author responses **addressed most of the raised concerns** and **raised the score to 6** with **the confidence score 5**.

- Efficiency and Applicability: The method's inference time (24.6 s) is faster than several diffusion-based methods. The peak VRAM consumption for the proposed method is 23169 MiB. (response to W1 & Q1)

- Theoretical Analysis: The approach decomposes features into semantic and style components; a cycle consistency step creates a semantic alignment domain by reducing style-related noise, which is validated empirically by resulting similarity maps focusing on true semantic regions. (response to W2)

- Ablation Experiments: Targeted ablation studies confirmed that the Sobel module acts as an effective structural safeguard to prevent excessive style injection, while the Gram module promotes style alignment; combining both achieves the best overall balance. (response to W3 & Q2)

- Additional Comparisons: Compared to Dreamstyler, the proposed method (CoCoDiff) shows more stable performance across FID, LPIPS, and CFSD metrics in terms of generation quality and structural preservation. (response to W4 & Q3)


### **Reviewer yGas (Most of the concerns have been addressed, last responded before the bug)**

- Reviewer yGas clearly claimed that **most of the concerns have been addressed** and expressed positive acknowledgment of our work and responses. The second-round comments were mainly aimed at improving the presentation, and we incorporated these suggestions in the revised manuscript. (We have already added before the bug.)

- More Comparison: The method (CoCoDiff) was compared against conventional patch-level methods like CNNMRF and Style-Swap, demonstrating superior or highly competitive performance across all metrics, indicating the benefit of its diffusion-based semantic alignment. We have incorporated both qualitative and quantitative comparisons with this research line in the main body of the manuscript as well as in the appendix. (response to W1)

- Correspondence Quality ($M$): The optimal feature space for style transfer is selected by evaluating the correspondence quality metric $M(t, l)$ for each timestep ($t$) and U-Net layer ($l$) using the PCK metric on the SPair-71k benchmark to balance semantic consistency and stylistic fidelity. (response to W2)

- Attention Mechanism Clarification : While modifying attention weights is common, this method explicitly establishes correspondence-aware attention paths and modulates attention only along these paths for more principled and targeted editing fidelity, with ablation confirming that joint Key (K) and Value (V) injection is essential for preserving both style and structure. (response to W3)

- Content and Style Loss: The content loss acts as a structural safeguard that stops the iterative optimization before the content structure becomes overly distorted, ensuring the dual constraints on style (Gram) and content (Sobel) jointly prevent both under and excessive stylization. (response to W4)

- User Study: We further explain the detail settings of User Study. (response to W5)

- Minor Details: All minor details and issues noted by the reviewer have been corrected in the revised manuscript. (response to W6)

---

> ### Author Response · Authors · 2025-12-02
>
> ### **Reviewer gHaT (no reply yet.)**
>
> - Correspondence Quality ($M$): The metric $M(t,l)$ quantifies the reliability of semantic correspondence within the feature space defined by timestep $t$ and U-Net layer $l$ by calculating the Percentage of Correct Keypoints (PCK) on benchmark datasets like SPair-71k; the optimal feature space  $(t^\*, l^\*)$ is chosen to maximize this PCK score. (response to W1)
>
> - Figure 2 Clarification: The Fine-grained Feature Matching (FFM) uses the style image ($I_{sty}$) for self-attention map extraction, and the Fitting cycle and Iterative Control (FIC) iteratively compares its intermediate output with the fixed reference images ($I_{sty}$ and $I_{con}$) to compute constraints; the accompanying figure will be revised for clarity. (response to W2)
>
> - Different Diffusion Models: The method (CoCoDiff) shows strong generalizability across different backbones (SD v1.5, v2.1, SDXL), but SD v1.4 was intentionally chosen for the main paper's results to ensure a fair and informative comparison with existing diffusion-based style transfer baselines. (response to W3)
>
> - Inference Time: The method's inference time (24.6 s) is competitive against most diffusion-based methods. (response to W4)
>
>
> ### **Reviewer hz3B (holds a positive attitude but no reply yet)**
>
> - Theoretical Analysis on Cycle Consistency: Direct feature similarity is biased by style-related residuals ($r_c, r_s$); the cycle consistency step constructs a "semantic alignment domain" by reducing style interference, allowing cosine similarity to more faithfully reflect the semantic inner product ($\langle u_c, u_s \rangle$), which is visually confirmed by resulting heatmaps concentrating on semantic regions. (response to W1 2)
>
> - Correspondence Quality ($M$): The metric $M(t,l)$ quantifies the semantic alignment accuracy by calculating the Percentage of Correct Keypoints (PCK) using features from timestep $t$ and layer $l$; the optimal feature space is selected by $\arg\max M(t,l)$ to ensure the feature vectors used for dense correspondence construction are reliable. (response to W2 Q1 and Q5)
>
> - Different Diffusion Models: While the method generalizes well across newer models (SD v1.5, v2.1, SDXL), SD v1.4 was deliberately chosen for the main results to ensure a fair and direct comparison with existing diffusion-based style transfer baselines. (response to Q3)
>
> - Time and Computational Costs: The method is efficient compared to most diffusion-based style transfer methods (24.6 s) and does not require extra training. Its peak GPU memory usage is 23169 MiB. (response to Q4)
>
>
> All of the above clarifications, new experiments, and rating updates were completed before post-discussion status of the reviews. Our general response about our core idea and our major improvements in rebuttal status is in the previous comment. Thank you again for your time and effort in carefully reviewing our submission.
>
> Authors

---

### Meta-Review · Area_Chair_nW3M · 2026-01-09

**Summary:**

This paper proposes a training-free diffusion-based framework for style transfer.  The proposed method establishes dense pixel-level semantic correspondences by mining intermediate diffusion features and introduces a cycle-consistency module to enforce structural and perceptual alignment.  The integration of semantic consistency into diffusion-based style transfer is interesting and the proposed method demonstrates good performance by mining intermediate diffusion features to construct dense pixel-level semantic correspondences between content and style images.  On the other hand, the reviewers are concerned with unclear details about the proposed method such as correspondence quality, analysis of cycle consistency or attention mechanism, insufficient evaluation including analysis of the obtained results, missing arguments on relevant prior work, and unclear presentation. In the rebuttal, the authors addressed the raised concerns adequately.

AC thinks that almost all the concerns were resolved by the rebuttal.  Reviewer1orb’s concern regarding presentation still remains: writing logic should be improved.  AC also points out an ethics issue: although user study with human subjects is conducted, no declaration of the approval of the study by the organization is addressed.  Taking all into account, AC thinks that this paper is deserved to be accepted, provided that the ethics issue is not critical.

**Reviewer Concerns:**

The main concerns are unclear detail about the proposed method, insufficient analysis of the obtained results, missing works in a relevant direction, and insufficient evaluation.  These concerns were basically resolved.  Room still remains to improve explanation in a logical and clear way, which does not degrade the contribution of this paper significantly.

**Reviewer Scores:**

Reviewer hz3B would keep his/her initial score or raise the score to 8 because his/her concerns are well addressed.  Reviewer gHaT would also keep his/her initial score or raise the score to 6 by the same reason as Reviewer hz3B.  Reviewer yGas would raise his/her score to 4 or 6 (probably 6) as most of his/her concerns were addressed.  Reviewer 1orb would raise his/her score to 6 as stated.

---

### Decision · Program_Chairs · 2026-01-26

Accept (Poster)